# Efficient Training of Boltzmann Generators Using Off-Policy Log-Dispersion Regularization

Henrik Schopmans [1] [2]   Christopher von Klitzing [1]   Pascal Friederich [1] [2]

## Abstract

Sampling from unnormalized probability densities is a central challenge in computational science. Boltzmann generators are generative models that enable independent sampling from the Boltzmann distribution of physical systems at a given temperature. However, their practical success depends on data-efficient training, as both simulation data and target energy evaluations are costly. To this end, we propose **off-policy log-dispersion regularization (LDR)**, a novel regularization framework that builds on a generalization of the log-variance objective. We apply LDR in the off-policy setting in combination with standard data-based training objectives, without requiring additional on-policy samples. LDR acts as a shape regularizer of the energy landscape by leveraging additional information in the form of target energy labels. The proposed regularization framework is broadly applicable, supporting unbiased or biased simulation datasets as well as purely variational training without access to target samples. Across all benchmarks, LDR improves both final performance and data efficiency, with sample efficiency gains of up to one order of magnitude.

## 1. Introduction

Sampling from complex, high-dimensional probability distributions is a central problem in computational physics, biology, and related areas. One such task is sampling from the Boltzmann distribution of physical systems, where $\tilde{p}_X(x) = \exp\left(\frac{-E(x)}{k_\mathrm{B}T}\right)$ is an unnormalized density, $x \in X$

is the configuration (e.g., particle positions in $\mathbb{R}^{3N}$), $E$ an energy function, $T$ the temperature, and $k_\mathrm{B}$ the Boltzmann constant. The normalized distribution is $p_X(x) = \tilde{p}_X(x)/\mathcal{Z}$ with $\mathcal{Z} = \int_{X_0} \tilde{p}_X(x)\,\mathrm{d}x$, where the integral is taken over $X_0$ corresponding to configurations with vanishing center of mass. The normalization constant $\mathcal{Z}$ is usually unknown and intractable.

Sampling from the Boltzmann distribution enables the exploration of typical configurations and estimating physically relevant expectations, e.g., in drug discovery where protein-ligand binding free energies require extensive exploration of thermodynamic ensembles (Hollingsworth & Dror, 2018). Classical sampling techniques such as molecular dynamics (MD) (Alder & Wainwright, 1959) and Markov chain Monte Carlo (MCMC) (Metropolis et al., 1953) are asymptotically correct but typically rely on long, correlated trajectories. As dimensionality and energetic complexity increase, these methods can become prohibitively expensive, particularly when energy evaluations are costly.

Generative modeling offers a complementary perspective by framing sampling as a one-shot generation problem. Boltzmann generators (Noé et al., 2019) enable independent sampling from equilibrium distributions at a given temperature, making them an attractive alternative or complement to trajectory-based methods. However, realizing a practical advantage crucially depends on data-efficient training procedures. Standard objectives for training Boltzmann generators typically rely on equilibrium samples from the target distribution (Tan et al., 2025b; Klein & Noé, 2024) or require extensive energy evaluations in the variational energy-based training setting (Schopmans & Friederich, 2025; von Klitzing et al., 2025), both of which may be limiting in realistic settings.

To address these limitations, we introduce **off-policy log-dispersion regularization (LDR)**, a general regularization framework for training Boltzmann generators more efficiently. LDR builds on a generalization of the log-variance objective, which has previously been used as a divergence for on-policy[1] variational training (Richter & Berner, 2023). In contrast, we apply LDR off-policy on fixed datasets

---

[1]Institute for Anthropomatics and Robotics, Karlsruhe Institute of Technology, Kaiserstr. 12, 76131 Karlsruhe, Germany [2]Institute of Nanotechnology, Karlsruhe Institute of Technology, Kaiserstr. 12, 76131 Karlsruhe, Germany. Correspondence to: Pascal Friederich <pascal.friederich@kit.edu>.

*Proceedings of the 43rd International Conference on Machine Learning*, Seoul, South Korea. PMLR 306, 2026. Copyright 2026 by the author(s).

[1]Using samples from the model distribution for training.

as a regularizer on top of standard data-based objectives. LDR uses target energy labels to regularize the shape of the learned energy landscape. Importantly, LDR can be evaluated over arbitrary reference distributions and can therefore leverage biased simulation data or auxiliary energy-labeled datasets that are otherwise difficult to incorporate.

The proposed framework applies to training on unbiased equilibrium data, biased simulation data, and even purely energy-based variational training without access to target samples. Across diverse benchmarks, we show that LDR significantly improves final model performance and data efficiency.

We summarize our contribution as follows:

- We adapt the log-variance objective from on-policy variational training to the off-policy setting, where it acts as a regularizer alongside standard data-based objectives for training on fixed, energy-labeled datasets.

- We generalize the log-variance objective to a family of log-dispersion objectives, including an L1 variant that is less sensitive to energy outliers and provides more stable optimization for Boltzmann generators.

- We show that log-dispersion regularization significantly increases final performance and data efficiency, both when training on unbiased and biased datasets.

- We show that log-dispersion regularization can also be adapted for energy-based variational training without target samples, improving the efficiency of the current state-of-the-art method, Constrained Mass Transport (CMT) (von Klitzing et al., 2025), by up to a factor of 10.

## 2. Related Work

Boltzmann generators were first introduced as an approach to obtain independent samples from Boltzmann distributions by learning an invertible generative model that supports exact likelihoods and importance reweighting (Noé et al., 2019). Closely related, Boltzmann emulators focus on learning data-driven models of approximate equilibrium ensembles, without exact reweighting guarantees (Jing et al., 2024; Lewis et al., 2025; Zheng et al., 2024).

Since their introduction, many works have studied Boltzmann generators trained from simulation data using standard data-based objectives, including discrete normalizing flows[2] (Midgley et al., 2023; Tan et al., 2025a; Rehman

---

[2]Here and throughout, "discrete" refers to normalizing flows parameterized as finite compositions of explicitly invertible transformations, such as coupling-based or autoregressively structured flows.

et al., 2025b), flow matching approaches (Klein et al., 2023; Rehman et al., 2025a), and diffusion-based Boltzmann generators (Zhang et al., 2025). Several works investigate transferability, training on some molecular systems and transferring to unseen ones to amortize simulation cost (Jing et al., 2022; Abdin & Kim, 2024; Klein & Noé, 2024; Tan et al., 2025b).

Beyond data-based training, Boltzmann generators can also be learned variationally using only target energy evaluations, without access to target samples. Variational methods based on discrete normalizing flows have advanced substantially in this regime (Stimper et al., 2022; Midgley et al., 2022; Schopmans & Friederich, 2025; von Klitzing et al., 2025), with recent approaches currently achieving strong performance for variational training in molecular systems (von Klitzing et al., 2025). Diffusion-based variational counterparts have also emerged (Liu et al., 2025; Choi et al., 2025; Kim et al., 2025), but have so far been less competitive on molecular benchmarks; notably, variational diffusion training on alanine dipeptide was only recently demonstrated (Liu et al., 2025), whereas flow-based approaches reached this earlier and have since scaled to larger systems (Midgley et al., 2022; von Klitzing et al., 2025).

Finally, several recent works explore training on data at elevated temperatures and annealing toward lower temperatures, combining generative modeling with tempering or progressive annealing schedules (Dibak et al., 2022; Wahl et al., 2025; Schopmans & Friederich, 2025; Rissanen et al., 2025; Akhound-Sadegh et al., 2025).

A key limitation across related work is data efficiency: in data-based approaches, the training signal typically comes from matching the sample distribution and does not directly leverage target energy values typically available through dataset generation. Likewise, in variational training, current state-of-the-art methods optimize objectives that do not include a direct training signal from target energy labels (Schopmans & Friederich, 2025; von Klitzing et al., 2025). Motivated by this gap, we introduce LDR, which improves data efficiency across a broad range of training settings. Similar to our work, Vaitl et al. (2024); Vaitl & Klein (2025) recently made an important step by improving the forward KL with a path-gradient formulation, yielding a lower-variance learning signal that incorporates gradients of the target density. Compared to this approach, LDR (i) is a general off-policy regularization framework that can be evaluated on arbitrary reference distributions (including biased or auxiliary energy-labeled datasets), and (ii) directly leverages target energies rather than target gradients. Finally, LDR is complementary to path gradients and can be used on top of the path-gradient forward KL as an additional energy-based regularizer.

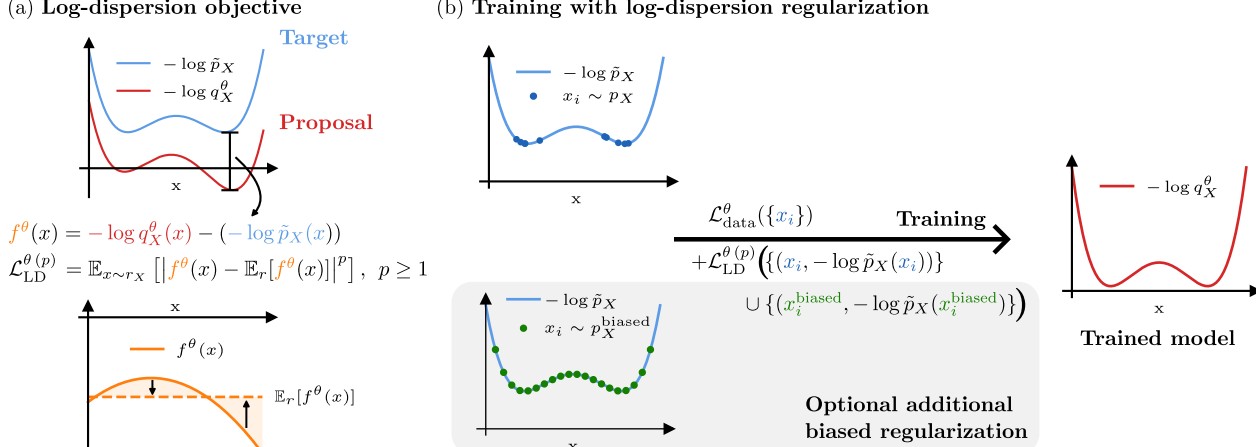

(a) **Log-dispersion objective**

(b) **Training with log-dispersion regularization**

*Figure 1.* (a) The log-dispersion objective minimizes the dispersion of $f^\theta(x) = -\log q_X^\theta(x) - (-\log \tilde{p}_X(x))$ around its mean, regularizing the shape of the proposal $q_X^\theta$ over the support of a reference distribution $r_X$. (b) When training off-policy, e.g., with a fixed dataset, log-dispersion alone is not a divergence because it does not constrain the proposal outside the support of $r_X$. We therefore use log-dispersion as a shape regularizer on top of data-based divergences to ensure global normalization. Since the reference distribution $r_X$ is flexible, LDR can be applied to target samples $x_i \sim p_X$ as well as biased samples $x_i^{\text{biased}} \sim p_X^{\text{biased}}$.

## 3. Method

To fit Boltzmann generators to a data distribution, they are typically trained with data-based objectives that do not use target energy values, even though these are usually available when constructing the dataset. Discrete normalizing flows are commonly trained via the forward KL divergence, continuous normalizing flows via flow matching, and diffusion models via score matching. While effective, these objectives rely solely on samples and ignore structure provided by the target energy landscape.

One way to incorporate target energy information is to directly compare the model log-density $\log q_X^\theta(x)$ to the unnormalized target log density $\log \tilde{p}_X(x)$. We define $f^\theta(x)$ as the difference of the negative log densities (unnormalized log importance weights),

$$
\begin{aligned}
f^\theta(x) &= -\log q_X^\theta(x) - (-\log \tilde{p}_X(x)) \\
&= -\log q_X^\theta(x) - \frac{E(x)}{k_{\text{B}}T}.
\end{aligned} \tag{1}
$$

At the optimum we have $q_X^\theta = p_X$, and thus $f^\theta(x) = \log \mathcal{Z} = \text{const.}$ (see Figure 1a for an illustration). To obtain this optimum, the log-variance objective (Richter et al., 2020) minimizes the variance of $f^\theta(x)$ under a reference distribution $r_X(x)$:

$$
\mathcal{L}_{\text{LV}}^\theta = \mathbb{E}_{r_X}\left[\left(f^\theta(x) - \mathbb{E}_{r_X}[f^\theta(x)]\right)^2\right] \tag{2}
$$

We generalize this log-variance objective to a family of log-dispersion objectives that minimize the dispersion of $f^\theta(x)$ around its mean,

$$
\mathcal{L}_{\text{LD}}^\theta = \mathbb{E}_{r_X}\left[\rho\left(f^\theta(x) - c^\theta\right)\right], \qquad c^\theta = \mathbb{E}_{r_X}\left[f^\theta(x)\right], \tag{3}
$$

$$
\rho(u) \geq 0, \qquad \rho(u) = 0 \iff u = 0.
$$

In our main experiments, we focus on $\rho(u) = |u|^p \ (p \geq 1)$, which yields $\mathcal{L}_{\text{LD}}^{\theta\,(p)} = \mathbb{E}_{r_X}\left[\left|f^\theta(x) - c^\theta\right|^p\right]$. We investigate LDR-L2 and LDR-L1, obtained by choosing $p = 2$ and $p = 1$, respectively. LDR-L2 recovers the log-variance objective and minimizes the second central moment (variance), while LDR-L1 minimizes the first absolute central moment. We hypothesize that LDR-L1 is less sensitive to outliers and numerically more stable, an important property for Boltzmann generators, where energy values can span multiple orders of magnitude. We compare additional log-dispersion variants in Appendix D.3. Some related diffusion sampling works replace the mean $c^\theta$ in Equation 3 with a learnable scalar parameter (trajectory balance objective) (Sendera et al., 2024). Here, we simply use the batch-wise mean and backpropagate through both $f^\theta(x)$ and $c^\theta$.

If $r_X(x)$ has full support, a member of the log-dispersion

family defines a divergence whose minimum is achieved if and only if $q_X^\theta = p_X$, assuming that $q_X^\theta$ is properly normalized. However, in practice, the configurations of a molecular system lie on a lower-dimensional manifold within the full space. Thus, using a fixed dataset as $r_X(x)$ typically yields a reference without full support, leaving the model unconstrained and arbitrarily normalized outside the data manifold (see Appendix D.2 for an empirical demonstration and Appendix A for a detailed theoretical analysis). Instead of using a fixed reference, related work on on-policy training with the log-variance objective chooses an adaptive on-policy reference distribution, $r_X(x) = q_X^\theta(x)$, but this requires repeated target energy evaluations and is prone to mode collapse in the finite-sample regime.

To avoid extra energy evaluations, we propose to deliberately demote the log-dispersion objective to a regularizer, combining it with a standard data-based objective:

$$\mathcal{L}^\theta = \lambda_{\text{data}} \mathcal{L}^\theta_{\text{data}} + \lambda_{\text{LD}} \mathcal{L}^\theta_{\text{LD}} . \qquad (4)$$

Figure 1b illustrates the training procedure using this log-dispersion regularization. The data-based term ensures correct normalization and convergence to the correct target distribution over the full space, while the log-dispersion term acts as a shape regularizer that aligns the proposal with the target energy landscape by minimizing dispersion in the log importance weights. This hybrid formulation provides an additional training signal through the incorporation of target energy labels. Importantly, LDR can be evaluated over arbitrary reference distributions and is not restricted to the target distribution itself. Furthermore, in contrast to on-policy training with log-dispersion objectives, our approach solely relies on a fixed dataset with target energy labels, as it is usually already available after performing unbiased or biased MD simulations. It does not have to use additional target energy evaluations during training.

## 4. Experiments

### 4.1. Experimental Setup

**Architecture** One key property of Boltzmann generators is the ability to asymptotically unbias the generated distribution using importance weights $w(x) = \frac{\tilde{p}_X(x)}{q_X^\theta(x)}$. For some observable $h(x)$ it holds (Martino et al., 2017; Noé et al., 2019)

$$\sum_{n=1}^N \frac{w(x_n)}{\sum_{i=1}^N w(x_i)} h(x_n) \xrightarrow[N \to \infty]{} \int h(x) p_X(x) \, \mathrm{d}x . \qquad (5)$$

While generative approaches such as continuous normal-

izing flows and diffusion models have shown strong performance, correcting their sampling bias via importance sampling is typically computationally expensive (Tan et al., 2025a; Klein & Noé, 2024). Recent works focused on cheaper unbiasing of continuous normalizing flows or diffusion models (Zhang et al., 2025; Peng & Gao, 2025), though no general solution is currently available. In contrast, discrete normalizing flows provide exact and inexpensive likelihood evaluations by construction. This property is especially critical in the context of Boltzmann generators, where unbiased estimates of thermodynamic quantities are required. For these reasons, we adopt discrete normalizing flows as the generative backbone in this work.

For individual molecular systems, normalizing flows defined on internal coordinate representations currently achieve state-of-the-art performance in both data-based and variational training regimes (Kim et al., 2024; von Klitzing et al., 2025). However, the choice of coordinate representation involves an inherent tradeoff: internal coordinates are system-specific and non-unique, which limits their transferability across different molecules. In contrast, Boltzmann generators operating directly on Cartesian coordinates offer a more flexible and transferable framework, but their performance still falls short of what can be achieved using internal coordinate representations (Klein & Noé, 2024; Tan et al., 2025b).

We use a two-step experimental setup. First, we evaluate LDR using normalizing flows in an internal coordinate representation (Sections 4.2, 4.3, and 4.4), which are substantially cheaper to train and evaluate and yield stronger final performance. We focus on controlled single-system benchmarks rather than transferability, where the non-uniqueness of internal coordinates would be a limiting factor. Second, to show that LDR is not tied to a particular coordinate choice, we report experiments in Cartesian coordinates in Section 4.5. For architecture details, see Appendix B.1.

**Target densities** To benchmark LDR, we first use alanine dipeptide ($d = 60$), which is a well-studied benchmark system in prior work (Dibak et al., 2022; Stimper et al., 2022; Midgley et al., 2022; Tan et al., 2025b). We further use the larger alanine hexapeptide ($d = 180$) (Schopmans & Friederich, 2025; von Klitzing et al., 2025) to test the scalability of our method. Both systems have complex metastable regions that occupy only a small part of the overall state space, making them well suited as challenging benchmarks.

**Metrics** As metrics, we first use the negative log-likelihood defined as NLL $= -\mathbb{E}_{x \sim p_X(x)} \left[ \log q_X^\theta(x) \right]$. The NLL is equivalent to the forward KL divergence up to an additive constant, and thus provides a reliable signal for mode collapse (Blessing et al., 2024) and model performance.

*Table 1.* Results for training on unbiased datasets from the target distribution. We report negative log-likelihood (NLL; lower is better) and effective sample size (ESS; higher is better), averaged over 4 independent experiments with standard deviations. For each number of samples, **bold** indicates metrics not significantly different from the best under uncorrected two-sided Welch's t-tests ($\alpha = 0.05$), capturing seed-level variability only. Right: ESS as a function of the number of training samples.

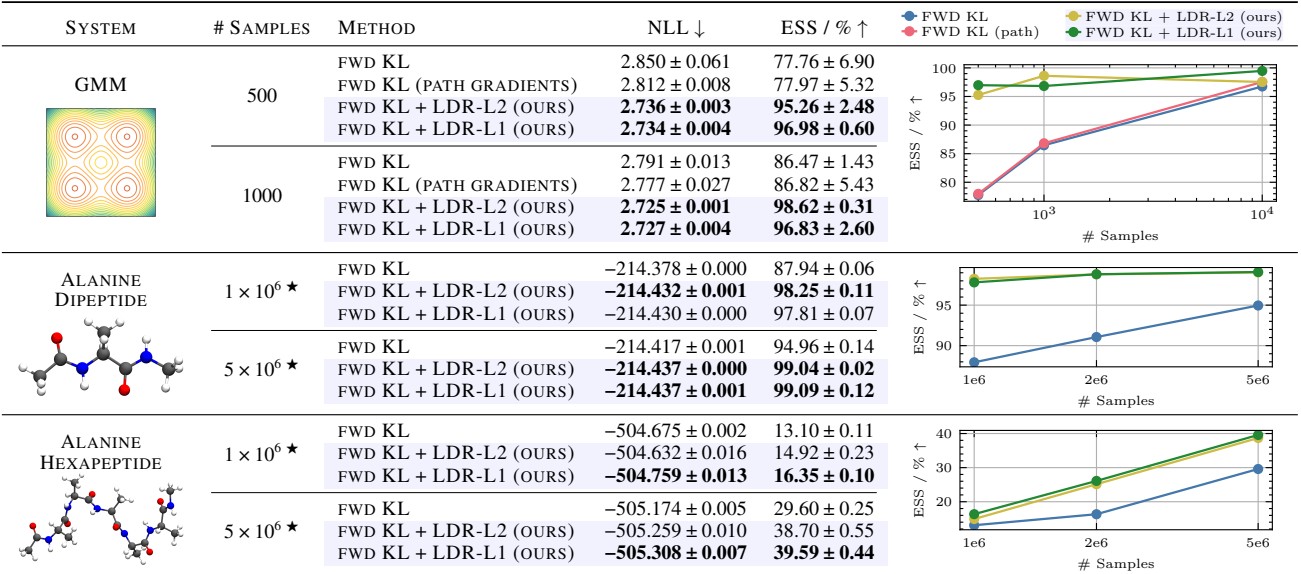

| SYSTEM | # SAMPLES | METHOD | NLL ↓ | ESS / % ↑ |
|---|---|---|---|---|
| GMM | 500 | FWD KL | $2.850 \pm 0.061$ | $77.76 \pm 6.90$ |
| | | FWD KL (PATH GRADIENTS) | $2.812 \pm 0.008$ | $77.97 \pm 5.32$ |
| | | FWD KL + LDR-L2 (OURS) | $\mathbf{2.736 \pm 0.003}$ | $95.26 \pm 2.48$ |
| | | FWD KL + LDR-L1 (OURS) | $\mathbf{2.734 \pm 0.004}$ | $\mathbf{96.98 \pm 0.60}$ |
| | 1000 | FWD KL | $2.791 \pm 0.013$ | $86.47 \pm 1.43$ |
| | | FWD KL (PATH GRADIENTS) | $2.777 \pm 0.027$ | $86.82 \pm 5.43$ |
| | | FWD KL + LDR-L2 (OURS) | $\mathbf{2.725 \pm 0.001}$ | $\mathbf{98.62 \pm 0.31}$ |
| | | FWD KL + LDR-L1 (OURS) | $\mathbf{2.727 \pm 0.004}$ | $96.83 \pm 2.60$ |
| ALANINE DIPEPTIDE | $1 \times 10^6$ ★ | FWD KL | $-214.378 \pm 0.000$ | $87.94 \pm 0.06$ |
| | | FWD KL + LDR-L2 (OURS) | $\mathbf{-214.432 \pm 0.001}$ | $\mathbf{98.25 \pm 0.11}$ |
| | | FWD KL + LDR-L1 (OURS) | $-214.430 \pm 0.000$ | $97.81 \pm 0.07$ |
| | $5 \times 10^6$ ★ | FWD KL | $-214.417 \pm 0.001$ | $94.96 \pm 0.14$ |
| | | FWD KL + LDR-L2 (OURS) | $\mathbf{-214.437 \pm 0.000}$ | $\mathbf{99.04 \pm 0.02}$ |
| | | FWD KL + LDR-L1 (OURS) | $\mathbf{-214.437 \pm 0.001}$ | $\mathbf{99.09 \pm 0.12}$ |
| ALANINE HEXAPEPTIDE | $1 \times 10^6$ ★ | FWD KL | $-504.675 \pm 0.002$ | $13.10 \pm 0.11$ |
| | | FWD KL + LDR-L2 (OURS) | $-504.632 \pm 0.016$ | $14.92 \pm 0.23$ |
| | | FWD KL + LDR-L1 (OURS) | $\mathbf{-504.759 \pm 0.013}$ | $\mathbf{16.35 \pm 0.10}$ |
| | $5 \times 10^6$ ★ | FWD KL | $-505.174 \pm 0.005$ | $29.60 \pm 0.25$ |
| | | FWD KL + LDR-L2 (OURS) | $-505.259 \pm 0.010$ | $38.70 \pm 0.55$ |
| | | FWD KL + LDR-L1 (OURS) | $\mathbf{-505.308 \pm 0.007}$ | $\mathbf{39.59 \pm 0.44}$ |

★ We note that we performed $> 1 \times 10^9$ target evaluations in the MD simulations used to construct the listed datasets via downsampling (Appendix B.4).

We additionally report the effective sample size (ESS), which quantifies how many independent draws from the target distribution $p_X$ would be required to attain the same estimator variance as that obtained using samples from the flow distribution $q_X^\theta$ (Martino et al., 2017). In practice, we compute the reverse ESS from flow samples (Martino et al., 2017). Since the reverse ESS is restricted to the support of $q_X^\theta$, a large ESS can occur under mode collapse; we therefore always assess it jointly with NLL.

We introduce and report additional metrics in Appendix B.5. In the main manuscript, consistent with recent work (Klein & Noé, 2024; Vaitl & Klein, 2025), we focus on NLL and ESS, which we found most reliable for assessing relative performance. However, NLL is not normalized, and its absolute scale is not directly interpretable, so even small differences can be significant. We therefore use NLL primarily to rank methods and refer to the metrics in the appendix when additional nuance is required.

We further refer to Figure 9 in Appendix F for a visualization of the 2D marginal densities of the main degrees of freedom of each system (Ramachandran and TICA plots).

### 4.2. Training on Unbiased Datasets

We start with the most straightforward setup, training on an unbiased dataset that follows the target distribution and includes target energy labels, $\mathcal{D}_n = \{(x_1, E(x_1)), \ldots, (x_n, E(x_n))\}$, $x_i \sim p_X$. Datasets used

to train Boltzmann generators are typically obtained from MD simulations, so target energy values already have to be computed during dataset creation. We can therefore apply energy regularization without extra cost, using $\mathcal{D}_n$ as the reference distribution of the LD objective.

To test the effect of LDR in different data regimes, we train on both alanine dipeptide and alanine hexapeptide using datasets of increasing size, i.e. $1 \times 10^6$, $2 \times 10^6$ and $5 \times 10^6$ samples. As a baseline, we use forward KL training without regularization, and compare it against models trained with LDR-L1 and LDR-L2 regularization.

In addition to the molecular benchmarks, we include a simple 2D Gaussian mixture model. Here, we provide the path-gradient forward KL (Vaitl et al., 2024) as an additional baseline, which uses gradients of the target density. For the molecular systems in internal coordinates, we do not include the path-gradient forward KL baseline, since we did not observe stable training in this setting; we include a detailed discussion and analysis in Appendix D.5.

**Results** Our results for training on unbiased datasets are summarized in Table 1 (full results can be found in Appendix E). Across all three tasks, we observe significant improvements in both NLL and ESS when using LDR compared to the baselines. For the Gaussian mixture task, LDR regularization allows stable training on only 500 data points, while the performance without LDR significantly degrades for this dataset size. For alanine dipeptide, LDR using

*Table 2.* Results for training on biased datasets with importance sampling (IS). We show results for pure data-based training with FWD KL, FWD KL + LDR on the IS samples only, and FWD KL + LDR on both the IS samples and the original biased dataset. We report negative log-likelihood (NLL; lower is better) and effective sample size (ESS; higher is better), averaged over 4 independent experiments with standard deviations. **Bold** indicates metrics not significantly different from the best under uncorrected two-sided Welch's t-tests ($\alpha = 0.05$), capturing seed-level variability only. Right: ESS as a function of the number of samples drawn for importance sampling.

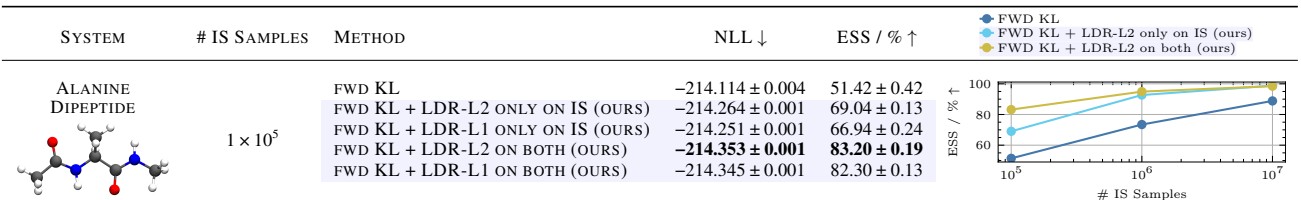

| SYSTEM | # IS SAMPLES | METHOD | NLL ↓ | ESS / % ↑ |
|---|---|---|---|---|
| ALANINE DIPEPTIDE | $1 \times 10^5$ | FWD KL | $-214.114 \pm 0.004$ | $51.42 \pm 0.42$ |
| | | FWD KL + LDR-L2 ONLY ON IS (OURS) | $-214.264 \pm 0.001$ | $69.04 \pm 0.13$ |
| | | FWD KL + LDR-L1 ONLY ON IS (OURS) | $-214.251 \pm 0.001$ | $66.94 \pm 0.24$ |
| | | FWD KL + LDR-L2 ON BOTH (OURS) | $\mathbf{-214.353 \pm 0.001}$ | $\mathbf{83.20 \pm 0.19}$ |
| | | FWD KL + LDR-L1 ON BOTH (OURS) | $-214.345 \pm 0.001$ | $82.30 \pm 0.13$ |

$1 \times 10^6$ samples outperforms training without LDR with $5 \times 10^6$ samples, indicating more than 5 times higher data efficiency. For alanine hexapeptide, we observe that LDR yields the largest improvements for $2 \times 10^6$ and $5 \times 10^6$ training samples (ESS plot on the right of Table 1). For $1 \times 10^6$ samples, the model likely remains too far from the target, leading to high-variance importance weights and a noisy LD objective, making it least effective in this regime. Moreover, LDR-L1 performs better than LDR-L2 for hexapeptide, consistent with the L1 objective being less sensitive to outliers. Appendix D.3 provides additional support for the improved stability of LDR-L1 through an analysis of gradient-norm variability during training.

### 4.3. Training on Biased Datasets

Unbiased molecular dynamics simulations become prohibitively expensive when the relevant configuration space is separated by large free-energy barriers or characterized by slow collective modes. In such regimes, trajectories can remain trapped in metastable regions for long times, making it infeasible to obtain sufficient equilibrium samples from $p_X$ within reasonable computational budgets.

By relaxing the requirement for unbiased samples, biased simulation techniques accelerate exploration by modifying the sampling dynamics, trading exact sampling from $p_X$ for improved state-space coverage (Abrams & Bussi, 2013). The resulting datasets are drawn from a biased distribution $p_X^{\text{biased}}$ but are often orders of magnitude cheaper to obtain than unbiased equilibrium data.

We train Boltzmann generators on biased simulation data using a two-stage procedure. First, we perform standard data-based training, using the forward KL on a biased dataset $\mathcal{D}_n^{\text{biased}} = \{x_i\}_{i=1}^n, x_i \sim p_X^{\text{biased}}$. This yields an initial proposal distribution $q_X^{\theta_1}$ that approximates the biased distribution. We use the same pre-trained model $q_X^{\theta_1}$ for all methods.

Second, we generate samples $x_j \sim q_X^{\theta_1}$ and compute importance weights $w(x_j) = \tilde{p}_X(x_j)/q_X^{\theta_1}(x_j)$. From this, we construct an approximate equilibrium dataset $\mathcal{D}_m^{\text{IS}}$ via

categorical resampling, i.e., by drawing samples with probabilities proportional to $w(x_j)$.

We then refine our initial biased proposal $q_X^{\theta_1}$ by training on $\mathcal{D}_m^{\text{IS}}$. Here, we compare standard forward KL training on $\mathcal{D}_m^{\text{IS}}$ with its LDR-L1-regularized and LDR-L2-regularized variants. This form of self-refinement has been previously used in other contexts (Schopmans & Friederich, 2025; Tan et al., 2025b). Since LDR can be evaluated over arbitrary reference distributions, we can use not only $\mathcal{D}_m^{\text{IS}}$ for LDR, but also a mixture of $\mathcal{D}_m^{\text{IS}}$ and the biased dataset $\mathcal{D}_n^{\text{biased}}$ (we use a $50 : 50$ mixture). This leverages additional training signal from the biased distribution, which is typically difficult to incorporate into Boltzmann generator training.

We note that the importance-sampling step requires sufficient overlap between $q_X^{\theta_1}$ and $p_X$; modes absent from the proposal cannot be recovered by resampling. If a (small) unbiased dataset is available, one can instead skip the importance sampling step and add arbitrary biased data to the reference distribution of the LDR objective, while using the unbiased data for the data-based objective. This allows incorporating biased distributions that do not cover all modes. However, we do not explore this option here, and focus on the importance sampling approach.

**Dataset** To test our methodology for training on biased datasets, we use short trajectories starting in the main modes of the energy landscape of alanine dipeptide. The short length of each trajectory leads to high correlations across trajectories, ultimately resulting in a heavily biased dataset of $1 \times 10^6$ samples. Details can be found in Appendix B.4.

In total, generating this biased dataset required only $1.1 \times 10^6$ target evaluations and can be performed in less than $200 \, \text{s}$ on a regular laptop. In contrast, typical MD simulations require $> 1 \times 10^8$ steps for high-quality unbiased datasets for alanine dipeptide (our ground truth MD simulation took $\sim 2$ days to simulate).

**Results** Results for training on the biased dataset of alanine dipeptide are summarized in Table 2 (full results can

*Table 3.* Results for variational training without access to target samples, using only target energy evaluations. We report negative log-likelihood (NLL; lower is better) and effective sample size (ESS; higher is better). We average over 4 independent experiments with standard deviations for alanine dipeptide, and 3 independent experiments for alanine hexapeptide to limit computational cost. For each number of target evaluations, **bold** indicates metrics not significantly different from the best under uncorrected two-sided Welch's t-tests ($\alpha = 0.05$), capturing seed-level variability only. Right: ESS as a function of the number of target energy evaluations.

| SYSTEM | METHOD | # TARGET EVALS ↓ | NLL ↓ | ESS / % ↑ | |
|---|---|---|---|---|---|
| ALANINE DIPEPTIDE | FAB | $2.13 \times 10^8$ | $-214.412 \pm 0.001$ | $94.95 \pm 0.12$ | |
| | TA-BG | $1 \times 10^8$ | $-214.419 \pm 0.002$ | $95.76 \pm 0.25$ | |
| | CMT | $1 \times 10^7$ | $-214.358 \pm 0.003$ | $85.20 \pm 0.47$ | |
| | CMT + LDR-L2 (OURS) | $1 \times 10^7$ | $\mathbf{-214.429 \pm 0.001}$ | $\mathbf{97.81 \pm 0.08}$ | |
| | CMT + LDR-L1 (OURS) | $1 \times 10^7$ | $\mathbf{-214.429 \pm 0.001}$ | $\mathbf{97.84 \pm 0.05}$ | |
| ALANINE HEXAPEPTIDE | FAB | $4.2 \times 10^8$ | $-504.355 \pm 0.019$ | $14.41 \pm 0.27$ | |
| | TA-BG | $4 \times 10^8$ | $-504.782 \pm 0.019$ | $18.66 \pm 0.16$ | |
| | CMT | $1 \times 10^8$ | $-503.190 \pm 0.051$ | $5.92 \pm 0.37$ | |
| | CMT + LDR-L2 (OURS) | $1 \times 10^8$ | $\mathbf{-504.801 \pm 0.008}$ | $\mathbf{31.47 \pm 0.71}$ | |
| | CMT + LDR-L1 (OURS) | $1 \times 10^8$ | $-504.721 \pm 0.010$ | $29.48 \pm 0.40$ | |

be found in Appendix E). Using only $1 \times 10^5$ importance samples, LDR applied on $\mathcal{D}_m^{\text{IS}}$ substantially improves performance compared to the non-regularized baseline. Applying regularization on both $\mathcal{D}_m^{\text{IS}}$ and the original biased dataset $\mathcal{D}_n^{\text{biased}}$ further improves performance, leading to a substantial gap. LDR effectively leverages information from the biased data distribution, leading to substantially improved refinement compared to standard importance-sampling-based self-refinement.

In total, LDR allows training of a well-performing Boltzmann generator using only $1 \times 10^5 + 1.1 \times 10^6$ (to construct the biased dataset) target evaluations. To achieve similar performance, non-regularized training requires more than 10 times more IS samples.

### 4.4. Variational Training without Target Data

Next, we show that regularizing data-based training also improves efficiency in the purely variational setting, where no target data (unbiased or biased) are available and training uses only target energy evaluations. Several recent methods train Boltzmann generators by annealing the model distribution toward the target through a sequence of intermediate distributions $q_i$ (Schopmans & Friederich, 2025; von Klitzing et al., 2025). To learn each intermediate distribution $q_i$, a buffer is built by reweighting model samples to the current intermediate distribution. Standard data-based training is then performed on each buffer until the next intermediate distribution is constructed, so we can readily apply LDR, using the buffer as the reference distribution.

Temperature-Annealed Boltzmann Generator (TA-BG) (Schopmans & Friederich, 2025) pre-trains with the reverse Kullback-Leibler (KL) divergence at an elevated temperature and subsequently anneals the model distribution using a number of intermediate temperatures $q_i \propto \tilde{p}_X^{\alpha_i}$. Con-

strained Mass Transport (CMT) (von Klitzing et al., 2025) extends this idea and uses a generalized annealing path of the form $q_i \propto q_0^{1-\lambda_i} (\tilde{p}_X^{\alpha_i})^{\lambda_i}$. The parameter schedule $(\lambda_i)_i$ is chosen adaptively based on a trust-region constraint that limits the KL divergence between subsequent intermediate distributions. This guarantees sufficient overlap between successive intermediates and improves performance compared to TA-BG. We provide a more detailed introduction to the annealing-based variational framework used in TA-BG and CMT in Appendix B.2. As an additional baseline, we include Flow Annealed Importance Sampling Bootstrap (FAB) (Midgley et al., 2022), which was the first to successfully learn alanine dipeptide variationally without mode collapse.

LDR can be applied to FAB, TA-BG, and CMT, as all use buffered off-policy training. Since CMT represents the current state of the art, we apply LDR only to CMT. For FAB and TA-BG, we run baseline experiments without LDR.

CMT provides a particularly suitable environment for applying LDR: by limiting the KL-divergence between intermediate distributions, it enforces sufficient overlap, which stabilizes the resulting log importance weights. This yields a lower-variance learning signal for log-dispersion than, e.g., in our experiments directly on unbiased datasets. We therefore view the combination of CMT and LDR as particularly promising for scaling equilibrium sampling to larger systems, although this remains an active area of research (Tan et al., 2025a;b) and is beyond the scope of this work.

**Results** Table 3 summarizes our results of applying LDR to variational training with CMT (full results can be found in Appendix E). LDR substantially improves the performance of CMT for all tested settings of total target evaluations. The gap becomes especially pronounced when reducing the number of target evaluations. On alanine dipeptide,

CMT + LDR requires only $1 \times 10^7$ target evaluations to achieve comparable performance to non-regularized CMT with $1 \times 10^8$ target evaluations (see Table 13). This indicates an approximately 10 times higher efficiency in terms of target evaluations.

CMT on alanine hexapeptide was originally reported with $4 \times 10^8$ target evaluations. In this setting, LDR substantially improves final performance, increasing ESS by more than 20 percentage points. When lowering the number of target evaluations to $1 \times 10^8$, vanilla CMT shows mode collapse (see Figure 9 in the appendix) and reaches an ESS of only 5.92 %. In contrast, LDR makes it possible to still achieve stable training without mode collapse in this setting, achieving an ESS of over 30 %. We further note that FAB and TA-BG perform considerably worse, even though they use $\geq 4$ times the number of target evaluations.

Overall, LDR allowed us to significantly reduce the number of target evaluations used in CMT while largely preserving final performance. We refer to Appendix D.4 for an ablation where we reduce the number of target evaluations even further than what we report in our main experiments.

### 4.5. Training Using Cartesian Coordinates

To show that the gains from LDR are not tied to internal coordinate representations, we additionally train Boltzmann generators directly on Cartesian coordinates. To match the data regime of related works, we use $1 \times 10^5$ training samples (Tan et al., 2025a). We use an autoregressive normalizing flow based on a transformer with causal masking (Zhai et al., 2025), adapted to Boltzmann generators by Tan et al. (2025a). Following Tan et al. (2025a), we incorporate the symmetries via data augmentation (random rotations and center-of-mass noise after centering), and thus train the flow in an augmented space. When estimating unbiased quantities via importance sampling, this requires a correction to map back to the target space; we propose an improved version of this correction compared to the one introduced by Tan et al. (2025a) (see Appendix C.1 for details).

We note that this discrete normalizing flow operating on Cartesian coordinates is substantially faster than continuous normalizing flow or diffusion-based counterparts used in related works. In Appendix B.1, we provide a detailed inference-speed comparison demonstrating that TarFlow achieves nearly a $4000\times$ higher sampling throughput than a representative continuous normalizing flow baseline.

**Results** Table 4 summarizes our results on Cartesian coordinates. First, we observe that our improved augmentation correction improves results from the original formulation (Tan et al., 2025a). Furthermore, LDR outperforms non-regularized forward KL by a substantial margin. Both LDR-L2 and LDR-L1 perform similarly well. We provide a de-

*Table 4.* Results for training on Cartesian coordinates using an unbiased dataset from the target distribution of alanine dipeptide. We average over 4 independent experiments with standard deviations. **Bold** indicates metrics not significantly different from the best under uncorrected two-sided Welch's t-tests ($\alpha = 0.05$), capturing seed-level variability only.

| METHOD | NLL $\downarrow$ | ESS / % $\uparrow$ |
|---|---|---|
| FWD KL (Tan et al., 2025a) | $-219.583 \pm 0.051$ | $19.40 \pm 0.80$ |
| FWD KL$^\star$ | $-219.583 \pm 0.050$ | $27.32 \pm 1.28$ |
| FWD KL$^\star$ + LDR-L2 (OURS) | $\mathbf{-219.815 \pm 0.034}$ | $\mathbf{35.89 \pm 1.69}$ |
| FWD KL$^\star$ + LDR-L1 (OURS) | $\mathbf{-219.800 \pm 0.044}$ | $\mathbf{35.75 \pm 1.63}$ |

$^\star$ Uses our improved augmentation correction, see Appendix C.1.

tailed discussion of these results in Appendix C.2. There, we also provide a comparison with the path-gradient version of the forward KL as an additional baseline (Vaitl et al., 2024). Our results show that LDR provides a strong additional training signal from energy labels also in the Cartesian-coordinate setting. While we focused on a controlled single-system benchmark, future work can explore training with LDR in the transferable setting (Tan et al., 2025b).

## 5. Discussion

While we focus on normalizing flows due to their accuracy and efficient importance sampling, LDR applies to any likelihood-based model class. It can also be applied to diffusion-based Boltzmann generators by reformulating Equation 3 over joint diffusion paths rather than terminal marginals, which upper-bounds the terminal objective (Sanokowski et al., 2025). While variational on-policy log-variance training for diffusion models is well-studied (Richter & Berner, 2023; Sendera et al., 2024), using it as off-policy regularization for training diffusion models has not been explored. However, we consider this outside the scope of the current manuscript.

Finally, we emphasize the strong dependence of final performance on dataset size observed across all experiments. The largest unbiased datasets used in this work ($5 \times 10^6$ samples) are considerably larger than those employed in related studies. We consistently observe substantial performance improvements when increasing dataset size beyond the regimes explored in prior work, both with and without LDR. This suggests that dataset size is a critical factor for Boltzmann generator training. Importantly, while LDR does not remove this dependence, it mitigates it by increasing data efficiency: models trained with LDR experience significantly improved metrics at smaller dataset sizes.

In the main experiments, we evaluated two variants of LDR, LDR-L1 and LDR-L2. Across all benchmarks, both variants yield comparable and substantial performance gains, indicating that the benefits of LDR are robust to the specific choice

of dispersion measure. We observe a slight advantage of LDR-L1 in the unbiased training of alanine hexapeptide, which we attribute to the heavy-tailed distribution of importance weights in this high-dimensional energy landscape. By penalizing absolute rather than squared deviations, LDR-L1 is less sensitive to outliers and therefore provides more stable optimization in this regime. Appendix D.3 contains additional experiments on other members of the log-dispersion family; we generally find that dispersion penalties with weaker tail sensitivity perform favorably. Overall, however, the performance difference between LDR-L1 and LDR-L2 remains small, suggesting that either variant is a reasonable default choice in practice.

**Limitations** A limitation of log-dispersion regularization is the introduction of an additional hyperparameter in the form of the relative loss weighting between LDR and the data-based objective. However, we observe only moderate sensitivity to this choice: as shown in Appendix D.1, performance remains stable over a range of loss weights.

A second limitation is that applying LDR with reference distributions far from the target may cause unstable training. Although this did not occur in our experiments, understanding LDR's behavior under increasingly biased references is an important direction for future work.

## 6. Conclusion

We introduced **off-policy log-dispersion regularization (LDR)**, a simple yet powerful framework for incorporating target energy information into the training of Boltzmann generators. By generalizing the log-variance objective and using it as an off-policy shape regularizer, LDR provides an additional training signal without requiring extra on-policy samples or additional target energy evaluations.

Across a wide range of settings, including training on unbiased equilibrium data, biased simulation datasets, and purely variational training without access to target samples, LDR consistently improves both final performance and data efficiency. In several benchmarks, these gains translate into order-of-magnitude reductions in the number of samples or target energy evaluations required to reach a given performance level. Importantly, the framework is broadly applicable and easy to integrate into existing pipelines.

## Reproducibility Statement

The source code to reproduce our experiments can be found on GitHub: `https://github.com/aimat-lab/Log-Dispersion-Regularization`. This repository also contains information on how to obtain the ground truth datasets necessary for training and evaluation.

## Acknowledgments

H.S. acknowledges financial support by the German Research Foundation (DFG) through the Research Training Group 2450 "Tailored Scale-Bridging Approaches to Computational Nanoscience". P.F. acknowledges funding from the Klaus Tschira Stiftung gGmbH (SIMPLAIX project) and the pilot program Core-Informatics of the Helmholtz Association (KiKIT project). Parts of this work were performed on the HoreKa supercomputer funded by the Ministry of Science, Research and the Arts Baden-Württemberg and by the Federal Ministry of Education and Research. The authors gratefully acknowledge the Gauss Centre for Supercomputing e.V. (www.gauss-centre.eu) for funding this project by providing computing time through the John von Neumann Institute for Computing (NIC) on the GCS Supercomputer JUWELS at Jülich Supercomputing Centre (JSC).

## Impact Statement

This paper presents work whose goal is to advance the field of machine learning. There are many potential societal consequences of our work, none of which we feel must be specifically highlighted here.

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

## A. Theory

We consider the log-dispersion objective

$$\mathcal{L}_{\mathrm{LD}}^{\theta}(r_X; \rho) = \mathbb{E}_{r_X}\left[\rho\left(f^{\theta}(x) - c^{\theta}\right)\right], \qquad c^{\theta} = \mathbb{E}_{r_X}\left[f^{\theta}(x)\right],$$

where the corresponding expectations are assumed to be well defined and $\rho$ satisfies

$$\rho(u) \geq 0, \qquad \rho(u) = 0 \iff u = 0.$$

### A.1. Optimum of Training with Log-Dispersion Objectives

**Proposition A.1** (Log-dispersion regularization as a divergence). *Let $r_X$ be a fixed reference distribution with full support on $X$, i.e., $r_X(x) > 0$ for all $x \in X$. Let $\tilde{p}_X(x) > 0$ be an unnormalized target density and let $q_X^{\theta}$ be a normalized model density with $q_X^{\theta}(x) > 0$ for all $x$. Then $\mathcal{L}_{\mathrm{LD}}^{\theta}(r_X; \rho) \geq 0$, and*

$$\mathcal{L}_{\mathrm{LD}}^{\theta}(r_X; \rho) = 0 \iff q_X^{\theta}(x) = \frac{\tilde{p}_X(x)}{\mathcal{Z}} = p_X(x).$$

*In particular, $\mathcal{L}_{\mathrm{LD}}^{\theta}(r_X; \rho)$ is a divergence whose unique minimum is attained at the target distribution.*

*Proof.* Nonnegativity is immediate since $\rho(u) \geq 0$.

If $\mathcal{L}_{\mathrm{LD}}^{\theta}(r_X; \rho) = 0$, then $\rho(f^{\theta}(x) - c^{\theta}) = 0$, which implies $f^{\theta}(x) = c^{\theta}$. Because $r_X$ has full support, this equality holds everywhere on $X$. Thus,

$$\log \frac{\tilde{p}_X(x)}{q_X^{\theta}(x)} = c^{\theta} \iff q_X^{\theta}(x) = e^{-c^{\theta}} \tilde{p}_X(x).$$

Normalization of $q_X^{\theta}$ yields

$$1 = \int q_X^{\theta}(x)\, dx = e^{-c^{\theta}} \int \tilde{p}_X(x)\, dx = e^{-c^{\theta}} \mathcal{Z},$$

and therefore $q_X^{\theta}(x) = \tilde{p}_X(x)/\mathcal{Z} = p_X(x)$.

Conversely, if $q_X^{\theta} = p_X$, then $f^{\theta}(x) = \log \mathcal{Z}$ is constant, and hence $\mathcal{L}_{\mathrm{LD}}^{\theta}(r_X; \rho) = 0$. $\qquad\square$

### A.2. Log-Dispersion Using Reference Distributions without Full Support

The divergence property of the log-dispersion objective established above relies critically on the assumption that the reference distribution $r_X$ has full support on $X$. In this work, however, we want to use fixed datasets, corresponding to a reference distribution whose support is restricted to a subset of configuration space. In this setting, the divergence property no longer holds.

**Proposition A.2** (LD without full support is not a divergence). *Let $r_X$ be a reference distribution whose support $\mathrm{supp}(r_X) \subsetneq X$ is a strict subset of the configuration space. Then $\mathcal{L}_{\mathrm{LD}}^{\theta}(r_X; \rho)$ is not a divergence on the space of normalized densities on $X$. In particular, there exist normalized densities $q_X^{\theta} \neq p_X$ such that*

$$\mathcal{L}_{\mathrm{LD}}^{\theta}(r_X; \rho) = 0.$$

*Proof.* If $\mathrm{supp}(r_X) \subsetneq X$, then the condition

$$\mathcal{L}_{\mathrm{LD}}^{\theta}(r_X; \rho) = 0$$

implies only that $f^{\theta}(x) = c^{\theta}$ holds $r_X$-almost surely, i.e., for all $x \in \mathrm{supp}(r_X)$. This yields

$$q_X^{\theta}(x) = e^{-c^{\theta}} \tilde{p}_X(x) \qquad \text{for } x \in \mathrm{supp}(r_X),$$

but places no constraint on $q_X^{\theta}(x)$ outside the support of $r_X$.

As a result, one can modify $q_X^{\theta}$ arbitrarily on $X \setminus \mathrm{supp}(r_X)$ while maintaining normalization, without affecting the value of $\mathcal{L}_{\mathrm{LD}}^{\theta}(r_X; \rho)$. Therefore, $\mathcal{L}_{\mathrm{LD}}^{\theta}(r_X; \rho) = 0$ does not imply $q_X^{\theta} = p_X$ on $X$, and log-dispersion fails to be a divergence in this case. Appendix D.2 illustrates this phenomenon empirically. $\qquad\square$

This situation arises naturally when $r_X$ corresponds to an empirical data distribution, e.g., obtained from molecular dynamics simulations, which typically cover only a low-dimensional manifold or a limited region of configuration space. In this regime, LD alone enforces agreement between $q_X^\theta$ and $p_X$ only on the data manifold, leaving the model unconstrained elsewhere and allowing incorrect normalization or spurious probability mass outside the data support. Furthermore, also for variational methods such as CMT, where off-policy training on fixed reweighted buffers is performed, the divergence property does not hold.

### A.3. Consistency of Log-Dispersion Regularization

We now show that combining the log-dispersion objective with a standard data-based divergence resolves this issue and restores the correct optimum even when the reference distribution lacks full support. We call this approach log-dispersion regularization.

Let $\mathcal{L}_{\text{data}}^\theta$ denote a data-based divergence whose unique minimum is attained at the target distribution $p_X$, such as the forward KL divergence, score matching, or flow matching, depending on the model class. We consider the combined objective

$$\mathcal{L}^\theta = \lambda_{\text{data}} \, \mathcal{L}_{\text{data}}^\theta + \lambda_{\text{LD}} \, \mathcal{L}_{\text{LD}}^\theta(r_X; \rho), \qquad \lambda_{\text{data}}, \lambda_{\text{LD}} > 0.$$

**Proposition A.3** (Consistency of log-dispersion regularization). *Assume that $\mathcal{L}_{\text{data}}^\theta \geq 0$ and that*

$$\mathcal{L}_{\text{data}}^\theta = 0 \quad \Longleftrightarrow \quad q_X^\theta = p_X.$$

*Then the combined objective $\mathcal{L}^\theta$ satisfies*

$$\mathcal{L}^\theta \geq 0, \qquad \mathcal{L}^\theta = 0 \quad \Longleftrightarrow \quad q_X^\theta = p_X,$$

*independently of whether the reference distribution used in $\mathcal{L}_{\text{LD}}^\theta(r_X; \rho)$ has full support.*

*Proof.* Nonnegativity follows immediately from nonnegativity of both terms and the assumption $\lambda_{\text{data}}, \lambda_{\text{LD}} > 0$.

If $\mathcal{L}^\theta = 0$, then necessarily $\mathcal{L}_{\text{data}}^\theta = 0$, which implies $q_X^\theta = p_X$ by assumption. Conversely, if $q_X^\theta = p_X$, then $\mathcal{L}_{\text{data}}^\theta = 0$ and $f^\theta(x) = \log \mathcal{Z}$ is constant, yielding $\mathcal{L}_{\text{LD}}^\theta(r_X; \rho) = 0$ for any reference distribution $r_X$. Thus $\mathcal{L}^\theta = 0$. $\qquad\square$

This result shows that log-dispersion can be safely employed as a regularization term when evaluated off-policy on a fixed dataset. The data-based objective ensures global correctness and proper normalization of the model distribution, while LDR provides an auxiliary training signal that aligns the learned density with the target energy landscape on the data manifold by reducing dispersion in the log importance weights. As a result, the combined objective preserves the true target distribution as its unique optimum, while benefiting from the additional structure encoded in the target energies.

### A.4. Gradients of Log-Dispersion Objectives

In this subsection, we specialize to the LDR-Lp objectives obtained by choosing $\rho(u) = |u|^p$ with $p \geq 1$. We analyze their gradients with respect to the model parameters $\theta$, with particular focus on the cases $p = 1$ and $p > 1$. Our goal is to understand the behavior of these gradients at the optimum $q_X^\theta = p_X$.

Recall that

$$f^\theta(x) = -\log q_X^\theta(x) - \frac{E(x)}{k_B T}, \qquad c^\theta = \mathbb{E}_{r_X}[f^\theta(x)],$$

and

$$\mathcal{L}_{\text{LD}}^{\theta\,(p)} = \mathbb{E}_{r_X}\left[\left|f^\theta(x) - c^\theta\right|^p\right].$$

**Gradient of the LDR-Lp objective**   Assuming sufficient regularity to interchange gradient and expectation, the gradient of $\mathcal{L}_{\text{LD}}^{\theta\,(p)}$ is given by

$$\nabla_\theta \mathcal{L}_{\text{LD}}^{\theta\,(p)} = p \, \mathbb{E}_{r_X}\left[\left|f^\theta(x) - c^\theta\right|^{p-1} \text{sign}\left(f^\theta(x) - c^\theta\right)\left(\nabla_\theta f^\theta(x) - \mathbb{E}_{r_X}[\nabla_\theta f^\theta(x)]\right)\right], \tag{6}$$

where $\text{sign}(u) = u/|u|$ for $u \neq 0$, and $\text{sign}(0)$ denotes the subdifferential $[-1, 1]$.

At the optimum $q_X^\theta = p_X$, we have

$$f^\theta(x) = \log \mathcal{Z} \quad \text{for all } x,$$

and therefore $f^\theta(x) - c^\theta = 0$ identically. The behavior of the gradient at this point depends critically on the choice of $p$.

**Case $p = 1$:** For $p = 1$, the LD objective reduces to

$$\mathcal{L}_{\text{LD}}^{\theta\,(1)} = \mathbb{E}_{r_X}\left[\left|f^\theta(x) - c^\theta\right|\right],$$

with gradient

$$\nabla_\theta \mathcal{L}_{\text{LD}}^{\theta\,(1)} = \mathbb{E}_{r_X}\left[\text{sign}\left(f^\theta(x) - c^\theta\right)\left(\nabla_\theta f^\theta(x) - \mathbb{E}_{r_X}[\nabla_\theta f^\theta(x)]\right)\right]. \tag{7}$$

At the exact optimum, $f^\theta(x) - c^\theta = 0$ for all $x$. However, the subdifferential of the absolute value at zero is the interval $[-1, 1]$, so the gradient is not uniquely defined. In the vicinity of the optimum, when $f^\theta(x) - c^\theta$ is small but nonzero, the gradient contributions in (7) remain of constant magnitude, independent of how close $q_X^\theta$ is to $p_X$. This contrasts with $p > 1$, where gradient contributions are smoothly damped as the optimum is approached (see below). Consequently, in stochastic optimization, where the model never exactly reaches $q_X^\theta = p_X$ and residual fluctuations persist, the L1 log-dispersion objective can induce persistent gradient noise near the optimum.

**Case $p > 1$:** For $p > 1$, the gradient in (6) contains the factor $\left|f^\theta(x) - c^\theta\right|^{p-1}$. At the optimum, where $f^\theta(x) - c^\theta = 0$ identically, this factor vanishes pointwise, since $p - 1 > 0$. Therefore,

$$\nabla_\theta \mathcal{L}_{\text{LD}}^{\theta\,(p)} = 0 \quad \text{for all } p > 1.$$

As the model distribution approaches the optimum $q_X^\theta = p_X$, the deviations $f^\theta(x) - c^\theta$ shrink, and the factor $|f^\theta(x) - c^\theta|^{p-1}$ in the gradient increasingly damps gradient contributions. At the optimum, this damping becomes exact, and the gradients of the LD term vanish identically. Consequently, once the model has reached the target distribution, the log-dispersion regularizer induces no further parameter updates.

**Implications** Although the $p = 1$ objective can be more robust to outliers due to its linear penalty, it may introduce persistent gradient noise near the optimum. In contrast, objectives with $p > 1$ combine vanishing gradients at convergence with increasing sensitivity to large deviations of $f^\theta(x)$ from its mean. This trade-off suggests a natural distinction between the cases $p = 1$ and $p > 1$ when employing log-dispersion regularization in practice.

## B. Experimental Setup

### B.1. Architecture

Here, we summarize the normalizing flow architectures used in this work. We further show the approximate inference time of each architecture in Table 5. To put the inference times into perspective, we also included a continuous normalizing flow, evaluated by exactly calculating the trace of the Jacobian using the codebase of Vaitl & Klein (2025).

*Table 5.* Approximate sampling speed for different architecture / system combinations, evaluated including the calculation of importance weights. For each architecture / system combination, we increased the batch size until either running out of memory or until the throughput did not increase further. Benchmarks were performed on a single NVIDIA A100 GPU (80 GB memory).

| SYSTEM | ARCHITECTURE | COORDINATE REPRESENTATION | SAMPLES / HOUR |
|---|---|---|---|
| ALANINE DIPEPTIDE | SPLINE FLOW | ICs | $\sim 2.8 \times 10^8$ |
| ALANINE HEXAPEPTIDE | SPLINE FLOW | ICs | $\sim 1.1 \times 10^8$ |
| ALANINE DIPEPTIDE | TARFLOW | Cartesian | $\sim 2.8 \times 10^7$ |
| ALANINE DIPEPTIDE | (Garcia Satorras et al., 2021) | Cartesian | $\sim 7300$ |

**GMM** To train on the GMM target density, we use a RealNVP architecture (Dinh et al., 2017) with affine coupling layers. The model consists of 15 coupling layers, each containing a conditioner network implemented as a fully-connected neural network with weight normalization and batch normalization. Each conditioner network has two hidden layers of 160 units with $\tanh$ activation functions. The coupling layers alternate their masking pattern to ensure all dimensions are transformed. The base distribution is a standard Gaussian $\mathcal{N}(0, I_2)$.

**Molecular systems in internal coordinates** The normalizing flow architecture follows established designs used in prior studies (Midgley et al., 2022; Schopmans & Friederich, 2025; 2024). Molecular conformations are represented in internal coordinates consisting of bond lengths, bond angles, and dihedral angles.

The model comprises 8 pairs of neural spline coupling layers based on monotonic rational-quadratic splines (Durkan et al., 2019). Each spline operates on the interval $[0, 1]$ and uses 8 bins. Within each pair of coupling layers, a random binary mask determines which dimensions are transformed and which are conditioned upon in the first layer, while the negated mask is applied in the second layer. Dihedral angle dimensions are handled using circular spline transformations (Rezende et al., 2020) to account for their periodic topology, and a fixed random periodic shift is applied after every coupling layer. The networks producing the spline parameters are fully connected neural networks with hidden layer sizes $[256, 256, 256, 256, 256]$ and ReLU nonlinearities. To encode periodicity, each dihedral angle $\psi_i$ is represented as $(\cos \psi_i, \sin \psi_i)$ when provided as input to the parameter networks.

The base distribution of the flow is chosen as a uniform distribution on $[0, 1]$ for dihedral angles, and a truncated Gaussian on $[0, 1]$ for bond lengths and bond angles, with mean $\mu = 0.5$ and standard deviation $\sigma = 0.1$.

Following Schopmans & Friederich (2025), all internal coordinates are mapped to the $[0, 1]$ domain required by the spline transformations. Dihedral angles are rescaled by division by $2\pi$. Bond lengths and angles are shifted and scaled according to $\eta_i' = (\eta_i - \eta_{i;\min})/\sigma + 0.5$, where $\eta_{i;\min}$ is taken from a minimum-energy configuration obtained via energy minimization. The scaling parameter $\sigma$ is set to 0.07 nm for bond lengths and to 0.5730 for bond angles.

The molecular systems considered admit two enantiomeric configurations corresponding to L- and R-chirality, whereas naturally occurring structures predominantly exhibit L-chirality. To restrict generated samples to the L-chiral manifold, the output ranges of the splines corresponding to the relevant dihedral angles are constrained as described in Schopmans & Friederich (2025). In addition, certain atoms or functional groups are formally permutation invariant in the force-field energy, but appear in a fixed ordering in molecular dynamics data. Analogously to the chirality constraints, the spline transformations are restricted such that only the permutations observed in the validation data can be generated (Schopmans & Friederich, 2025).

To implement the normalizing flow models on internal coordinate representations, we used the *bgflow* (Noé, 2024) and *nflows* (Durkan, Conor et al., 2020) libraries.

**Molecular systems in Cartesian coordinates** For our experiments on Cartesian coordinates, we leverage TarFlow (Zhai et al., 2025), a block-wise autoregressive normalizing flow based on a transformer architecture with causal masking. We closely follow the hyperparameters introduced by Tan et al. (2025a). We use 4 transformation blocks, each with 4 attention layers and 256 channels. Cartesian coordinates are transformed block-wise, treating the three Cartesian coordinates of each atom as one block.

### B.2. Variational Training Based on Annealing Paths

This section provides a short overview of the annealing-based variational sampling approaches CMT (von Klitzing et al., 2025) and TA-BG (Schopmans & Friederich, 2025).

Let $p_X : \mathbb{R}^d \to \mathbb{R}^+$ be a probability density function known up to its normalization constant $\mathcal{Z}$, i.e.,

$$p_X(x) = \frac{\tilde{p}_X(x)}{\mathcal{Z}}, \quad \text{with} \quad \mathcal{Z} = \int_{\mathbb{R}^d} \tilde{p}_X(x)\mathrm{d}x.$$

A common strategy to generate samples $x \sim p_X(x)$ is to approximate $p_X$ with a parameterized variational model $q_X^\theta \in \mathcal{Q}_\theta \subset \mathcal{P}(\mathbb{R}^d)$ by minimizing the reverse KL divergence,

$$\min_\theta D_{\mathrm{KL}}(q_X^\theta \| p_X). \tag{8}$$

While this approach works well for simple target distributions, more complex targets often lead to mode collapse, where some modes of $p_X$ are poorly represented or entirely missed by the model.

**Annealing paths**   One strategy to mitigate this issue is the use of variational annealing paths, which interpolate between an initial model distribution $q_0$ and the target distribution $p_X$ through a sequence of intermediate densities $(q_i)_{i=1}^K$, with $q_K = p_X$. Recent work on constrained variational objectives by von Klitzing et al. (2025) (CMT) establishes a connection between a family of constrained variational objectives and common annealing paths. We therefore consider the geometric-tempered annealing path

$$q_i \propto q_0^{1-\lambda_i}(p_X^{1/T_i})^{\lambda_i}, \quad \lambda_i \in \mathbb{R}_{\geq 0}, \ T_i \in [1, \infty), \ i = 1 \ldots K,$$

proposed by CMT (von Klitzing et al., 2025), which generalizes the widely used geometric annealing path

$$q_i \propto q_0^{1-\lambda_i} p_X^{\lambda_i}, \quad \lambda_i \in \mathbb{R}_{\geq 0}, \ i = 1 \ldots K,$$

and the temperature annealing path

$$q_i \propto p_X^{1/T_i}, \quad T_i \in [1, \infty), \ i = 1 \ldots K,$$

used by TA-BG (Schopmans & Friederich, 2025).

**Annealing schedules**   TA-BG first pre-trains the model at an elevated temperature using the reverse KL objective. Since modes are more interconnected at high temperatures, this helps avoid mode collapse. TA-BG subsequently anneals the model distribution, starting from the high-temperature distribution $q_0$, using a manually chosen geometric temperature annealing schedule.

In contrast, CMT (von Klitzing et al., 2025) skips the pre-training phase and directly starts with an uninformed prior distribution $q_0$. von Klitzing et al. (2025) then derive a geometric-tempered annealing path and its adaptive schedule by analytically solving the constrained variational objective

$$q_{i+1} = \underset{q_X \in \mathcal{P}(\mathbb{R}^d)}{\arg\min} \ D_{\mathrm{KL}}(q_X \| p_X) \quad \text{s.t.} \quad D_{\mathrm{KL}}(q_X \| q_i) \leq \varepsilon_{\mathrm{tr}}, \quad \mathcal{H}(q_i) - \mathcal{H}(q_X) \leq \varepsilon_{\mathrm{ent}}, \quad \int q_X(x)\,\mathrm{d}x = 1$$

for general probability measures $q \in \mathcal{P}(\mathbb{R}^d)$. While this approach can adaptively choose both $(\lambda_i)_i$ (trust-region constraint) and $(T_i)_i$ (entropy constraint), we found it simpler to select the temperature schedule manually, as in TA-BG, while still enforcing the trust-region constraint. Using a manual temperature schedule thus yields the objective

$$q_{i+1} = \underset{q_X \in \mathcal{P}(\mathbb{R}^d)}{\arg\min} \ D_{\mathrm{KL}}(q_X \| p_X^{1/T_{i+1}}) \quad \text{s.t.} \quad D_{\mathrm{KL}}(q_X \| q_i) \leq \varepsilon_{\mathrm{tr}}, \quad \int q_X(x)\,\mathrm{d}x = 1.$$

---

**Algorithm 1** Annealing algorithm

---

**Require:** Initial density $q_0$, target density $\tilde{p}_X$, divergence $D$, approximation family $\mathcal{Q}_\theta$
  **for** $i \leftarrow 1, \ldots, K$ **do**
    Initialize buffer $\mathcal{B}_i$ with samples from current model $q_X^\theta$
    Prepare for next intermediate target $q_i$ (e.g., choose $\lambda_i$ adaptively, compute importance weights, ...)
    **for** $k \leftarrow 1, \ldots, L$ **do**
      Update $q_X^\theta$ by performing gradient descent on $D(q_i, q_X^\theta)$ using the buffer $\mathcal{B}_i$
    **end for**
  **end for**
  **return** $q_X^\theta \approx p_X$

---

**Algorithm**   Given an annealing path and schedule, the original objective in Equation (8) breaks down into a sequence of simpler variational objectives

$$\min_\theta D(q_i, q_X^\theta), \qquad i = 1 \ldots K,$$

where $D$ denotes a statistical divergence. Both CMT and TA-BG employ the importance-weighted forward Kullback-Leibler divergence.

*Table 6.* Summary of the molecular systems considered in this study, along with the associated force-field parametrizations. To be consistent with prior work, we consider two different force-field variants for alanine dipeptide. The first is used for our experiments in internal coordinates, while the latter is used for the experiments in Cartesian coordinates.

| NAME | NO. ATOMS | SEQUENCE | FORCE FIELD | CONSTRAINTS |
|---|---|---|---|---|
| ALANINE DIPEPTIDE | 22 | ACE-ALA-NME | Amber ff96 with OBC1 implicit solvation | NONE |
| ALANINE DIPEPTIDE | 22 | ACE-ALA-NME | Amber ff99SB-ILDN with Amber99 OBC implicit solvation | NONE |
| ALANINE HEXAPEPTIDE | 62 | ACE-5·ALA-NME | Amber ff99SB-ILDN with Amber99 OBC implicit solvation | HYDROGEN BOND LENGTHS |

Both variational approaches follow a nested optimization structure, consisting of an outer and an inner loop. In each outer iteration, a buffer of samples is generated and then used off-policy to update the model parameters during the inner loop. We can thus readily apply LDR in this setting. An abstract version of the annealing algorithm is provided in Algorithm 1.

### B.3. Target Densities

**GMM target** We use a Gaussian mixture model (GMM) in $d = 2$ dimensions, closely following the system introduced by Vaitl et al. (2024). The target distribution is defined as an equally-weighted mixture of $2^d$ Gaussian components arranged on a regular grid. The component means are positioned at the vertices of a hypercube with coordinates $\mu_i \in \{-1, +1\}^d$, resulting in $2^2 = 4$ modes located at $(\pm 1, \pm 1)$. Each component is an independent isotropic Gaussian with standard deviation $\sigma = 0.5$.

**Molecular systems** All our experiments on molecular systems were performed at 300 K. An overview of the molecular systems investigated in this work, together with the corresponding force-field parametrizations, is provided in Table 6. All energy evaluations used for model training were carried out with OpenMM version 8.0.0 (Eastman et al., 2024) on the CPU platform, employing 18 parallel workers.

In line with previous studies (Midgley et al., 2022; Schopmans & Friederich, 2025; von Klitzing et al., 2025), we employ a regularized energy formulation to mitigate excessively large van der Waals contributions arising from atomic overlaps:

$$E_{\text{reg}}(E) = \begin{cases} E, & \text{if } E \leq E_{\text{high}}, \\ \log(E - E_{\text{high}} + 1) + E_{\text{high}}, & \text{if } E_{\text{high}} < E \leq E_{\text{max}}, \\ \log(E_{\text{max}} - E_{\text{high}} + 1) + E_{\text{high}}, & \text{if } E > E_{\text{max}}. \end{cases} \quad (9)$$

We choose $E_{\text{high}} = 1 \times 10^8$ and $E_{\text{max}} = 1 \times 10^{20}$, following the values reported by Midgley et al. (2022).

*Table 7.* Summary of the number of trainable parameters for each architecture / system combination.

| SYSTEM | ARCHITECTURE | COORDINATE REPRESENTATION | NO. PARAMETERS |
|---|---|---|---|
| GMM | REALNVP | - | 403 230 |
| ALANINE DIPEPTIDE | SPLINE FLOW | ICs | 7 421 512 |
| ALANINE HEXAPEPTIDE | SPLINE FLOW | ICs | 12 124 616 |
| ALANINE DIPEPTIDE | TARFLOW | Cartesian | 12 668 952 |

### B.4. Datasets

We performed extensive molecular dynamics simulations to obtain high-quality ground truth datasets for both training and evaluation. Our simulation protocol is similar to that reported by Schopmans & Friederich (2025) and von Klitzing et al. (2025).

For all molecular systems, we performed two independent simulations: the first was used to build a training dataset with $5 \times 10^6$ samples and a validation dataset of $1 \times 10^6$ samples, the second to build a test dataset with $1 \times 10^7$ samples. The validation dataset was used to optimize hyperparameters, and the test dataset to report the final metrics.

**Alanine dipeptide**    For alanine dipeptide, we performed molecular dynamics simulations with the OpenMM integrator *LangevinMiddle* and a time step of 1 fs. We first equilibrated for 200 ns, followed by the production simulation of 5 µs.

**Alanine hexapeptide**    For alanine hexapeptide, we performed replica-exchange molecular dynamics (Sugita & Okamoto, 1999) simulations at temperatures [300.0 K, 332.27 K, 368.01 K, 407.60 K, 451.44 K, 500.0 K] with the OpenMM integrator *LangevinMiddle* and a time step of 1 fs. We first equilibrated each replica independently for 200 ns without exchanges, then equilibrated for 200 ns with exchanges, and subsequently performed the production simulation of 2 µs per replica. To build our datasets, the data of the 300.0 K replica was used.

**Biased dataset for alanine dipeptide**    To produce the biased dataset for alanine dipeptide (see Section 4.3 of the main text), we used 4 starting configurations located in the main minima of the energy surface of alanine dipeptide, see Figure 2. In each configuration, we started 20 MD trajectories at 300 K, each with a length of 13750 steps and time step 1 fs using a *LangevinMiddle* integrator. We discarded the first 1250 steps of each trajectory. Since we recorded every step of the trajectories, this resulted in a biased dataset with $1 \times 10^6$ samples.

This construction uses prior knowledge of the main modes of alanine dipeptide through the choice of starting configurations. Consequently, the total number of target energy evaluations used in our biased experiments is not directly comparable to the number of evaluations required for the unbiased experiments in Section 4.2; these costs should therefore be interpreted separately.

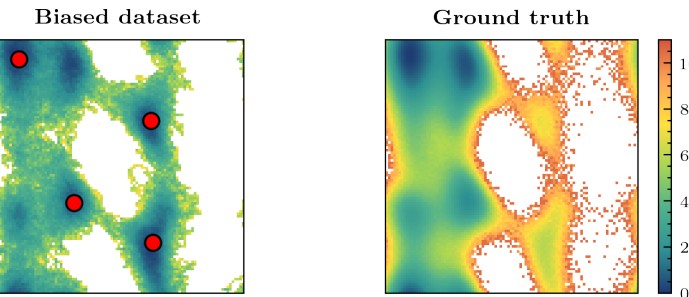

*Figure 2.* Ramachandran plot of the biased training dataset for alanine dipeptide. The four starting configurations of the trajectories are labeled in red. Due to the short length of the trajectories, the high-energy metastable states on the right side are oversampled compared to the ground truth.

## B.5. Metrics

As discussed in the main part of our manuscript, we use the negative log-likelihood (NLL) and ESS as the main metrics of our experiments. We also introduce several additional metrics below. All metrics were estimated using $1 \times 10^7$ samples from the model and ground truth (when needed). For the experiments in Cartesian coordinates with TarFlow, we used $1 \times 10^6$ samples for evaluation due to the reduced inference speed.

**NLL**    The NLL is defined as

$$\text{NLL} = -\mathbb{E}_{x \sim p_X(x)} \left[ \log q_X^\theta(x) \right] . \tag{10}$$

For the experiments on Cartesian coordinates with data augmentation, we evaluate the NLL on the augmented distribution $p_{\text{aug}}(x)$, as this is ultimately the one learned by the model (see Appendix C.1 for details).

**ESS**     The ESS can be calculated using the following equation (Martino et al., 2017; Midgley et al., 2023):

$$\text{ESS} = \frac{n_{\text{e,rv}}}{N} = \frac{1}{N \sum_{i=1}^{N} \bar{w}(x_i)^2} \tag{11}$$

$$\text{with} \quad x_i \sim q_X^{\theta}(x_i), \quad \bar{w}(x_i) = \frac{w(x_i)}{\sum_{j=1}^{N} w(x_j)}, \quad w(x) = \frac{\tilde{p}_X(x)}{q_X^{\theta}(x)}$$

To calculate the ESS, the top $0.01\%$ of importance weights were truncated to their minimum value within this subset, following previous work (Midgley et al., 2022; Schopmans & Friederich, 2025; von Klitzing et al., 2025). This clipping is performed to remove outliers due to model instabilities from the importance weights. The effective sample size was computed using the regularized energy function (Equation 9).

**Ramachandran metrics**     Ramachandran plots visualize the two-dimensional log-density of the joint distribution of backbone dihedral angle pairs $(\phi, \psi)$ in a peptide. They capture the dominant conformational degrees of freedom of molecular systems and are particularly sensitive to mode collapse and missing high-energy regions.

To quantitatively assess deviations between ground-truth and model-generated Ramachandran plots, we follow previous work (Midgley et al., 2022; Schopmans & Friederich, 2025; von Klitzing et al., 2025) and compute the forward Kullback-Leibler divergence between the discretized ground truth Ramachandran distribution and the model distribution, which we call **RAM KL**. Both distributions were estimated using $100 \times 100$ bins over the full dihedral angle range. Since alanine hexapeptide has 5 pairs of backbone dihedral angles, we average over the 5 corresponding **RAM KL** metrics.

In addition, we report a reweighted variant of this metric (**RAM KL w. RW**), where model samples are first reweighted to the target distribution using importance weights before constructing the Ramachandran histogram. To improve the numerical stability of the reweighted estimate and to suppress the influence of rare outliers caused by model instabilities, the same clipping procedure as used for the ESS computation is applied: the top $0.01\%$ of importance weights are clipped to the minimum value within this subset.

**TICA metrics**     Time-lagged independent component analysis (TICA) provides a low-dimensional representation of the slow collective degrees of freedom of molecular systems. Distributions in TICA space are therefore sensitive to deficiencies in the learned slow dynamics and to mode collapse in kinetically relevant regions.

To quantitatively assess deviations between ground-truth and model-generated distributions in TICA space, we compute the forward Kullback–Leibler divergence between the discretized ground truth TICA distribution and the model distribution, which we call **TICA KL**. Both distributions were estimated using $100 \times 100$ bins over the TICA ranges determined from the ground-truth samples and using the first two TICA components.

In addition, we report a reweighted variant of this metric (**TICA KL w. RW**), where model samples are first reweighted to the target distribution using importance weights before constructing the TICA histogram, using the same importance weight clipping as used for **RAM KL w. RW**.

**Metrics based on energy distribution**     In line with related work (Tan et al., 2025a), we further measure the discrepancy between generated and reference target energy distributions by computing the 2-Wasserstein distance between their one-dimensional distributions over potential energy values (estimated from samples). Lower $\mathcal{E}\text{-}\mathcal{W}_2$ indicates a closer match of the generated energy histogram to the reference. Since this metric is very sensitive to outliers, in line with related work, we only report this metric after categorical resampling of the flow sample distribution according to the importance weights. For our experiments in internal coordinates, we use $1 \times 10^7$ flow samples and resample to $1 \times 10^7$ samples. For our experiments in Cartesian coordinates, we use $1 \times 10^6$ flow samples and resample to $1 \times 10^6$ samples. We note that, in contrast to Wasserstein distances in $d > 1$, they can be efficiently calculated even for very large sample sizes in 1D.

We emphasize that this histogram energy metric compares only a one-dimensional marginal distribution, and can therefore be less informative than metrics such as ESS, which more directly reflects the quality of the reweighting and the match to the target distribution. Moreover, across all our experiments the reweighted energy histograms are already very close to the target; hence, $\mathcal{E}\text{-}\mathcal{W}_2$ mainly captures small residual fluctuations, and we caution against placing too much weight on small differences in this metric.

**Further metrics used in related work**  Several related works report the torus Wasserstein distance ($\mathbb{T}$-$\mathcal{W}_2$) on backbone dihedral angles and the Wasserstein distance in the TICA projection space, TICA-$\mathcal{W}_2$ (Tan et al., 2025a;b; von Klitzing et al., 2025; Rehman et al., 2025a;b).

This metric is typically estimated using $10^4$ samples (some more recent works use up to $2.5 \times 10^5$ samples), where the sample-based dihedral angle distribution can still be a very coarse proxy for the underlying target distribution; for example, even in the two-dimensional Ramachandran marginals of alanine dipeptide, the qualitative appearance changes drastically as the number of reference samples increases from $10^4$ to $10^7$ (Figure 3). Especially to properly resolve the high-energy metastable region on the right side of the Ramachandran, which is crucial to be sensitive with respect to mode collapse, a large number of samples is necessary. Increasing the number of samples to match the $10^7$ samples we use for the metrics **RAM KL** and **TICA KL** would make Wasserstein-based evaluation prohibitively expensive, particularly when considering higher-dimensional torus products for longer peptides such as alanine hexapeptide.

For these reasons, we focus on our KL-based metrics, which can be estimated reliably at very large sample sizes and which we found to yield more stable and reliable comparisons.

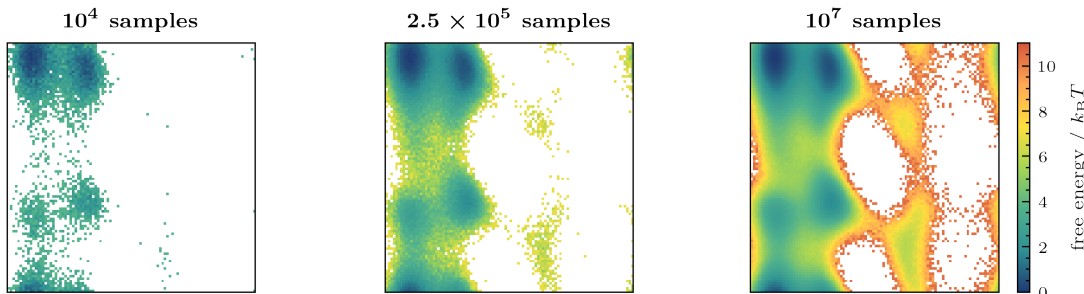

*Figure 3.* Ramachandran plots for alanine dipeptide, using randomly chosen $1 \times 10^4$ samples (left), $2.5 \times 10^5$ (middle), and $1 \times 10^7$ samples (right) from the ground truth dataset.

### B.6. Hyperparameters

**General**  All experiments were performed using the Adam optimizer (Kingma & Ba, 2017) and were implemented in *PyTorch* (Paszke et al., 2019). All experiments included a cosine annealing learning rate scheduler with a single cycle.

Initial learning rates were tuned separately for each experiment, using a logarithmically spaced grid search. We first evaluated a coarse grid spanning several orders of magnitude, and subsequently refined the search around the best-performing region using a finer grid of the form $\{\ldots, 10^{-4}, 3.2 \times 10^{-5}, 10^{-5}, \ldots\}$.

All data-based training methods (unbiased and biased) used a batch size of 1024.

The total number of gradient descent steps for each method can be found in Table 8.

**LDR**  To report best-case performance with LDR, we tuned $\lambda_{\text{data}}$ over the grid $\{0.1, 0.3, 0.5, 0.7, \ldots\}$ while fixing $\lambda_{\text{LD}} = 1$ for the experiments in internal coordinates. We emphasize, however, that LDR performs well across a range of choices of $\lambda_{\text{data}}$, as shown in Figures 4 and 5. For the experiments in Cartesian coordinates, we set $\lambda_{\text{data}} = 1$ and tuned $\lambda_{\text{LD}}$ over the grid $\{0.1, 0.3, 0.5, 0.7, \ldots\}$.

In the data-based LDR experiments, we additionally warmed up the effective regularization weight at the beginning of training. This makes the early optimization steps dominated by the data-based objective before LDR is gradually switched on. Specifically, we multiplied the final value of $\lambda_{\text{LD}}$ by a cosine ramp,

$$\lambda_{\text{LD}}(t) = \lambda_{\text{LD}} \begin{cases} \frac{1}{2}(1 - \cos(\pi t / T_{\text{warm}})), & t < T_{\text{warm}}, \\ 1, & t \geq T_{\text{warm}}. \end{cases} \tag{12}$$

For the unbiased LDR experiments in internal coordinates, we used $T_{\text{warm}} = 2 \times 10^4$ optimization steps. For the biased

LDR experiments on alanine dipeptide, we used $T_{\text{warm}} = 5 \times 10^4$ optimization steps. For the unbiased LDR experiments in Cartesian coordinates, we used $T_{\text{warm}} = 1 \times 10^5$ optimization steps. For CMT + LDR, we did not schedule $\lambda_{\text{LD}}$ and instead used the final weight throughout training.

**Biased experiments**   For the importance sampling step in the biased experiments, we categorically resampled the importance-weighted dataset to a new dataset of $1 \times 10^7$ samples.

**CMT**   We used a batch size of 1000 for alanine dipeptide and 2000 for alanine hexapeptide. Gradient norm clipping was applied with a threshold of 100. At the start of training, the learning rate was linearly warmed up over the first 1000 steps. The trust-region bound was set to 0.3 for all experiments.

For alanine dipeptide, we used 200 outer annealing steps, while 400 annealing steps were used for alanine hexapeptide. To compute the number of samples for the buffer in each annealing step, we uniformly distributed the total number of target energy evaluations over the number of annealing steps.

As described in Section B.2, we did not use the entropy constraint introduced by von Klitzing et al. (2025). Instead, we chose a manual geometric temperature schedule that geometrically anneals the temperature from 1200 K to 300 K over the first half of the total number of gradient descent steps. With this fixed schedule, we observed CMT to achieve comparable results as when using an entropy constraint. In this setup, CMT is very similar to TA-BG with an additional trust-region constraint to improve overlap of consecutive distributions.

**TA-BG**   For our experiments with TA-BG, we started from the exact hyperparameters specified by Schopmans & Friederich (2025). We refer to (Schopmans & Friederich, 2025) for details. Reverse KL pre-training was performed at 1200 K, from where a geometric annealing path to 300 K was followed. To ensure a fair comparison, we matched the number of gradient descent steps used in TA-BG to those used in CMT (see Table 8).

**FAB**   For our experiments with FAB, we used the exact hyperparameters specified by von Klitzing et al. (2025), without additional tuning.

### B.7. Training Times

We summarize the total training time (excluding evaluation) observed on an NVIDIA A100 GPU (40 GB memory) for each method in Table 8. We emphasize that LDR does not change the required training time and can be added with no extra cost.

*Table 8.* Summary of the number of gradient descent steps and the total training time (excluding evaluation) for each method.

| SYSTEM | ARCHITECTURE (REP.) | METHOD | NO. GRADIENT STEPS | TRAINING TIME |
|---|---|---|---|---|
| GMM | REALNVP | UNBIASED DATASET | 10 000 | $\sim$ 4 min |
| ALANINE DIPEPTIDE | SPLINE FLOW (IC) | UNBIASED DATASET | 250 000 | $\sim$ 11.8 h |
| ALANINE HEXAPEPTIDE | SPLINE FLOW (IC) | UNBIASED DATASET | 250 000 | $\sim$ 17.3 h |
| ALANINE DIPEPTIDE | SPLINE FLOW (IC) | BIASED DATASET | 250 000 | $\sim$ 12.6 h |
| ALANINE DIPEPTIDE | SPLINE FLOW (IC) | CMT ($1 \times 10^7$ EVALS) | 400 000 | $\sim$ 17.9 h |
| ALANINE HEXAPEPTIDE | SPLINE FLOW (IC) | CMT ($1 \times 10^8$ EVALS) | 800 000 | $\sim$ 53.0 h |
| ALANINE DIPEPTIDE | TARFLOW (CART.) | UNBIASED DATASET | 400 000 | $\sim$ 18.6 h |

## C. Training on Cartesian Coordinates

### C.1. Augmentation

Directly enforcing translational and rotational invariance of the target distribution within a discrete normalizing flow architecture is challenging due to inherent architectural constraints of normalizing flows. As a result, Tan et al. (2025a) adopt an alternative strategy and train a non-invariant flow, while accounting for these symmetries through data augmentation as follows:

1. Let $X = \mathbb{R}^{N \times 3}$. For a configuration $x \in X$, write $x_i \in \mathbb{R}^3$ for its $i$-th row, i.e., the Cartesian position of particle $i$, and

define

$$\bar{x} := \frac{1}{N} \sum_{i=1}^{N} x_i, \qquad x^\circ := x - \mathbf{1}_N \bar{x}^\top,$$

where $\mathbf{1}_N \in \mathbb{R}^N$ denotes the column vector of ones. Thus $\mathbf{1}_N \bar{x}^\top \in \mathbb{R}^{N \times 3}$ repeats the center of mass in every row.

2. Define the augmentation map
$$s : X_0 \times \mathbb{R}^3 \to X, \qquad s(x^\circ, t) = x^\circ + \mathbf{1}_N t^\top.$$

3. Sample an independent translation:
$$t \sim \mathcal{N}(0, \sigma_t^2 I_3).$$

4. Apply the augmentation (push-forward):
$$x_{\mathrm{aug}} = s(x^\circ, t) = x^\circ + \mathbf{1}_N t^\top$$

**Note:** While molecular dynamics (MD) simulations, in principle, explore all possible molecular rotations due to the rotation-invariance of $p_X$, in practice, we additionally apply randomly sampled rotations during training, which increases sample diversity and robustness.

Adding noise to the center of mass is crucial. Without this noise, the data distribution lies on a $3N - 3$-dimensional submanifold of $\mathbb{R}^{3N}$. As a result, any mapping from a full-dimensional base distribution would require a singular Jacobian, violating the invertibility assumptions of normalizing flows.

In the following, we derive the push-forward density $p_{\mathrm{aug}}(x)$ of the augmentation scheme described above.

**Lemma C.1** (Center-of-mass augmentation). *Let $p_{X_0}$ be a probability density on the centered subspace*

$$X_0 := \{z \in X : \bar{z} = 0\},$$

*assume $p_{X_0}$ is rotation invariant, i.e., $p_{X_0}(zR) = p_{X_0}(z)$ for all $R \in \mathrm{SO}(3)$, and let $t \sim \mathcal{N}(0, \sigma_t^2 I_3)$.*

*The augmented density $p_{\mathrm{aug}}(x)$ on the full space $X$, given by the push-forward using the map $s$, is, up to normalization:*

$$p_{\mathrm{aug}}(x) \propto p_{X_0}(x^\circ) \mathcal{N}(\bar{x} \mid 0, \sigma_t^2 I_3).$$

*Proof.* We begin with the joint density of the independent variables $x^\circ \in X_0, t \in \mathbb{R}^3$:

$$p(x^\circ, t) = p_{X_0}(x^\circ) \mathcal{N}(t \mid 0, \sigma_t^2 I_3).$$

A change-of-variables of this joint distribution using the map $s(x^\circ, t) = x$ leads to an augmented distribution on the full space, given by

$$p_{\mathrm{aug}}(x) = p\big(s^{-1}(x)\big) \cdot |\det J|^{-1} \quad \text{with} \quad J = \frac{\partial s}{\partial(x^\circ, t)}$$

Substituting the joint density expression and the transformation, we obtain:

$$p_{\mathrm{aug}}(x) \propto p_{X_0}\big(x - \mathbf{1}_N \bar{x}^\top\big) \mathcal{N}(\bar{x} \mid 0, \sigma_t^2 I_3) \cdot |\det J|^{-1}$$

Since $s$ is a linear transformation, its Jacobian determinant is constant. This leaves us with

$$p_{\mathrm{aug}}(x) \propto p_{X_0}(x^\circ) \mathcal{N}(\bar{x} \mid 0, \sigma_t^2 I_3),$$

which completes the proof. □

**Augmented loss functions**  We can now express the augmented versions of both the forward KL and the LD objective using the non-augmented target distribution $p_{X_0}(x)$ as

$$\min_\theta D_{\mathrm{KL}}(p_{\mathrm{aug}} \| q_X^\theta) = \min_\theta -\mathbb{E}_{p_{\mathrm{aug}}(x)}[\log q_X^\theta(x)] = \min_\theta \;\; -\mathbb{E}_{\substack{x^\circ \sim p_{X_0} \\ t \sim \mathcal{N}(0, \sigma_t^2 I_3)}} [\log q_X^\theta(x^\circ + \mathbf{1}_N t^\top)]$$

and

$$\min_\theta \mathcal{L}_{\mathrm{LD}}^\theta = \min_\theta \;\; \mathbb{E}_{p_{\mathrm{aug}}(x)}\left[\rho\left(f^\theta(x) - \mathbb{E}_{p_{\mathrm{aug}}(x)}[f^\theta(x)]\right)\right]$$

$$= \min_\theta \;\; \mathbb{E}_{\substack{x^\circ \sim p_{X_0} \\ t \sim \mathcal{N}(0, \sigma_t^2 I_3)}}\left[\rho\left(f^\theta(x^\circ, t) - \mathbb{E}_{\substack{x^\circ \sim p_{X_0} \\ t \sim \mathcal{N}(0, \sigma_t^2 I_3)}} [f^\theta(x^\circ, t)]\right)\right],$$

with

$$f^\theta(x^\circ, t) = \log \tilde{p}_{X_0}(x^\circ) + \log \mathcal{N}(t | 0, \sigma_t^2 I_3) - \log q_X^\theta(x^\circ + \mathbf{1}_N t^\top).$$

**Importance sampling correction**  Using the loss functions above, the normalizing flow learns to model the augmented distribution $p_{\mathrm{aug}}(x)$, rather than the (non-augmented) target density $p_{X_0}$. To account for this during importance sampling with respect to $p_{X_0}$, Tan et al. (2025a) introduced the following correction of the proposal density:

$$\log q_X^{\theta c}(x) = \log q_X^\theta(x) + \frac{\|\bar{x}\|^2}{2\sigma_t^2} - \log\left[\frac{\|\bar{x}\|^2}{\sqrt{2}\sigma_t^3 \Gamma\left(\frac{3}{2}\right)}\right] \tag{13}$$

This correction effectively removes a $\chi_3$ distribution from the proposal density.

However, as shown above, the augmented density that the normalizing flow learns factorizes as

$$p_{\mathrm{aug}}(x) \propto p_{X_0}(x^\circ) \mathcal{N}(\bar{x} \mid 0, \sigma_t^2 I_3). \tag{14}$$

Consequently, the discrepancy between the learned proposal and the target density arises solely from the additional Gaussian factor in the augmented coordinates. A natural correction, therefore, consists of explicitly removing this Gaussian contribution from the proposal density, yielding

$$\log q_X^{\theta c}(x) = \log q_X^\theta(x) - \log \mathcal{N}(\bar{x} \mid 0, \sigma_t^2 I_3) \tag{15}$$

This recovers a proposal density consistent with importance sampling toward $p_{X_0}$. Our results show empirically that this correction yields improved results compared to the one proposed by Tan et al. (2025a) (Table 4 and Table 14).

### C.2. Extended Discussion of Results

As mentioned in the main text, we also used the path-gradient estimator for the forward KL objective as a baseline in our Cartesian-coordinate experiments (see Table 14); we provide additional details here. For this method, we relied on the "plug-and-play" implementation provided by Vaitl et al. (2024). In our setup, this implementation was approximately $4\times$ slower per optimizer step than the other training objectives considered. To match the overall computational budget, we therefore ran path-gradient forward KL for only $100\,000$ optimizer steps, whereas the other methods were trained for $400\,000$ steps.

Despite this constraint, we do observe the characteristic low-variance behavior of path-gradient forward KL: when measured as a function of the number of gradient updates, it tends to converge faster than the alternatives. However, this advantage

does not translate to improved performance as a function of wall-clock time in our experiments, due to its substantially higher per-step cost.

We note that Vaitl & Klein (2025) addressed a closely related issue in the context of continuous normalizing flows—where the computational gap is much more pronounced—by first pre-training with flow matching and then fine-tuning with path gradients. However, to keep the experimental setup simple, we did not include pre-training and fine-tuning pipelines in our setting.

Finally, we emphasize that path-gradient forward KL is not a direct competitor to LDR. Conceptually, path-gradient forward KL regularizes training through the inclusion of target *gradients*, whereas LDR regularizes through the inclusion of target *energy values*. These mechanisms are complementary and can be readily combined in future work.

# D. Additional Experiments and Discussion

## D.1. Sensitivity to Loss Weights

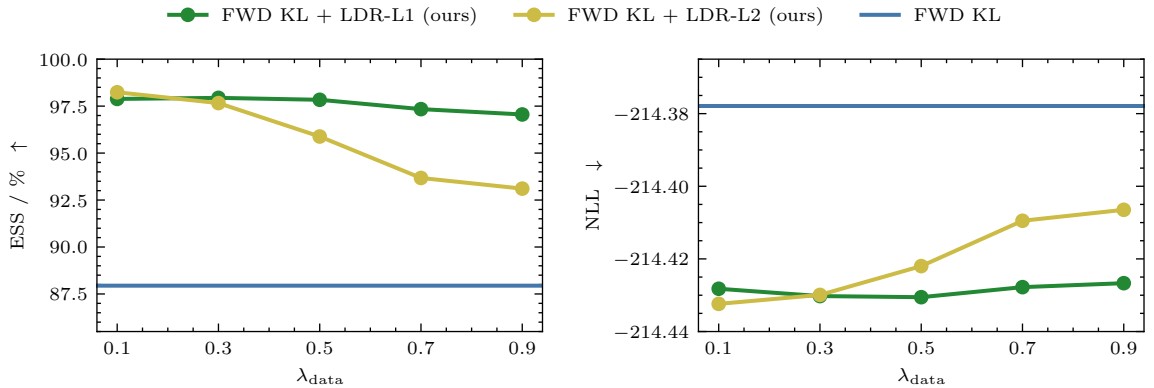

*Figure 4.* Final ESS and NLL as a function of the loss weight $\lambda_{\text{data}}$ for unbiased training on alanine dipeptide using $1 \times 10^6$ samples.

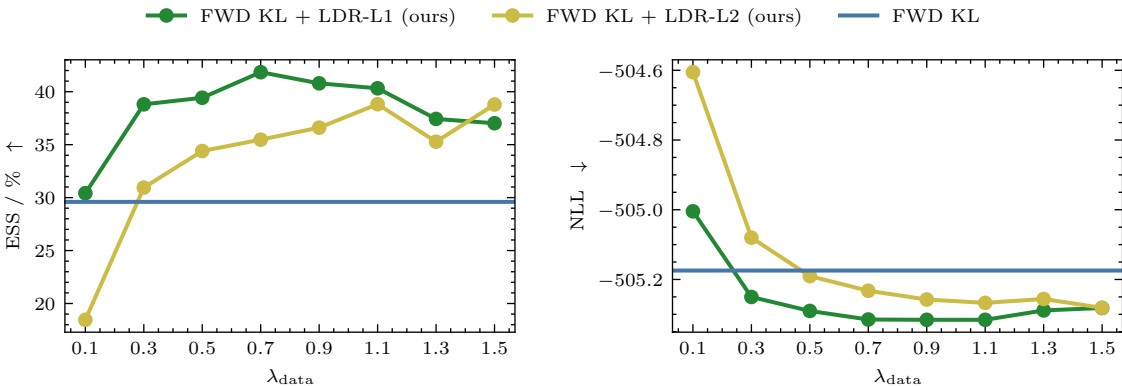

*Figure 5.* Final ESS and NLL as a function of the loss weight $\lambda_{\text{data}}$ for unbiased training on alanine hexapeptide using $5 \times 10^6$ samples.

Compared to standard data-based training, our method introduces an additional hyperparameter that scales the two loss components relative to each other. We visualize the sensitivity of the final model performance to the choice of the hyperparameter $\lambda_{\text{data}}$ in Figures 4 and 5. For each chosen $\lambda_{\text{data}}$, we tune the learning rate separately to remove confounding effects caused by non-optimal settings. As one can see, both LDR-L1 and LDR-L2 outperform the baseline without

regularization across a large range of chosen $\lambda_{\text{data}}$. Furthermore, LDR-L1 appears somewhat more stable with respect to $\lambda_{\text{data}}$ compared to LDR-L2.

## D.2. Training without a Data-Based Objective

As outlined in Appendix A, LDR needs to be combined with a divergence that ensures correct normalization of the proposal. Figure 6 illustrates what happens when only LDR is used for training on an unbiased dataset. While the proposal matches the target where the dataset has support, it is unconstrained outside of the data manifold.

The same problem occurs when running CMT (as well as TA-BG) with only LDR. Both methods build buffers by reweighting model samples to an intermediate target distribution. Since the LDR objective is evaluated over this reweighted fixed dataset while training on each buffer, the same problem exists; the objective does not constrain the model density outside the data manifold.

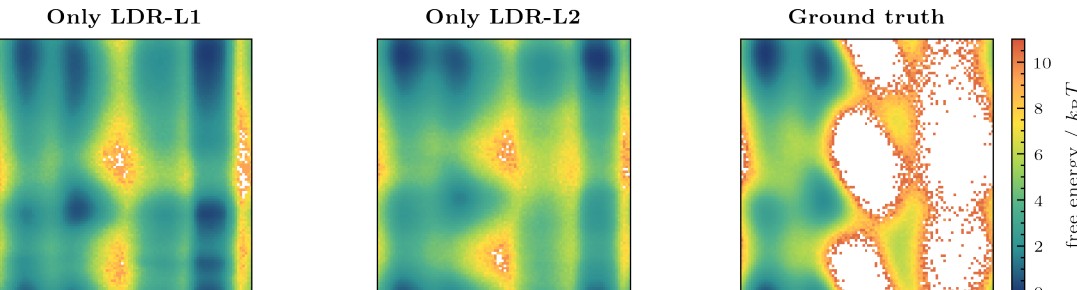

*Figure 6.* Ramachandran obtained for alanine dipeptide when training with only LDR-L1 or only LDR-L2, without an additional data-based divergence.

## D.3. Additional Log-Dispersion Variants

In the main text of this work, we focused on the LDR-L1 and LDR-L2 variants from the log-dispersion family. In Table 9, we compare the performance of additional variants when training on unbiased data for alanine hexapeptide ($1 \times 10^6$ samples). This additionally includes LDR-L1.5 ($p = 1.5$), LDR-L3 ($p = 3$), and LDR-Huber. For LDR-Huber, we use

$$\rho(u) = \begin{cases} u^2, & |u| \le \delta, \\ 2\delta|u| - \delta^2, & |u| > \delta, \end{cases}$$

with $\delta = 1.0$ in our experiments.

*Table 9.* Comparison of additional log-dispersion variants for unbiased training on alanine hexapeptide using $1 \times 10^6$ samples. We average over 4 independent experiments with standard deviations. **Bold** indicates metrics not significantly different from the best under uncorrected two-sided Welch's t-tests ($\alpha = 0.05$), capturing seed-level variability only.

| METHOD | NLL $\downarrow$ | ESS / % $\uparrow$ |
|---|---|---|
| FWD KL | $-504.675 \pm 0.002$ | $13.10 \pm 0.11$ |
| FWD KL + LDR-L1 (OURS) | $\mathbf{-504.759 \pm 0.013}$ | $\mathbf{16.35 \pm 0.10}$ |
| FWD KL + LDR-L1.5 (OURS) | $\mathbf{-504.750 \pm 0.007}$ | $\mathbf{16.74 \pm 0.20}$ |
| FWD KL + LDR-L2 (OURS) | $-504.632 \pm 0.016$ | $14.92 \pm 0.23$ |
| FWD KL + LDR-L3 (OURS) | $-504.425 \pm 0.020$ | $8.95 \pm 0.26$ |
| FWD KL + LDR-HUBER (OURS) | $\mathbf{-504.752 \pm 0.016}$ | $\mathbf{16.88 \pm 0.40}$ |

Table 9 shows that LDR-L1, LDR-L1.5, and LDR-Huber achieve similar performance on this task. In contrast, LDR-L2 performs worse than variants with weaker tail sensitivity, and LDR-L3 degrades performance further. This supports the interpretation from the main text that penalties with weaker sensitivity to large residuals can provide more stable optimization in the heavy-tailed importance-weight regime of alanine hexapeptide.

This trend is also visible in Figure 7, where we plot the variance of the L2 gradient norm inside a sliding window during training. LDR-L1 exhibits the lowest gradient-norm variance, while LDR-L2 and LDR-L3 show larger variation, indicating less stable training dynamics.

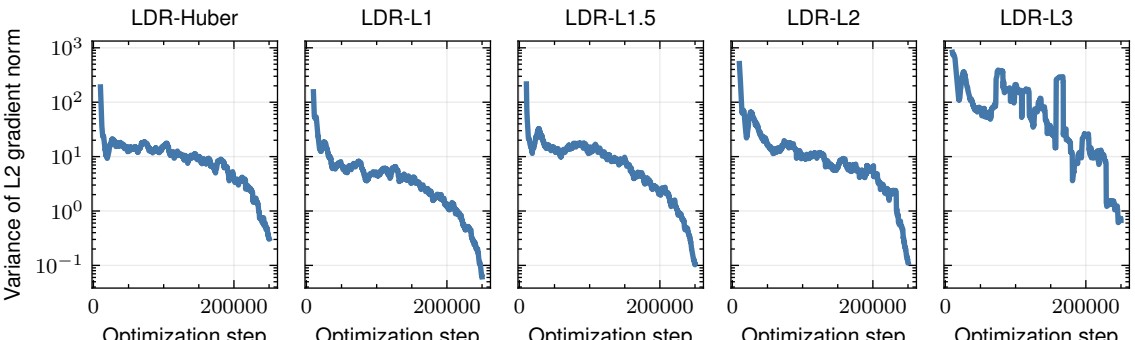

*Figure 7.* Variance of L2 gradient norm inside a sliding window of 10 000 optimization steps during training. We compare multiple variants from the log-dispersion family for training on unbiased data for alanine hexapeptide with $1 \times 10^6$ samples.

### D.4. Pushing the Limits of CMT

The LDR objective introduces an additional training signal, enabling effective variational training with substantially fewer gradient descent steps and total target energy evaluations. While our main experiments use $1 \times 10^7$ target evaluations as the lowest setting of CMT for alanine dipeptide, this ablation further reduces the evaluation budget.

Figure 8 reports the final performance of CMT, CMT + LDR-L1, and CMT + LDR-L2 under a linear scaling of both the number of gradient descent steps and the buffer size per annealing step, while fixing the number of annealing steps at 200. Although performance degrades as the training budget is reduced, CMT with log-dispersion regularization is significantly more robust to fewer gradient descent steps and target energy evaluations than non-regularized CMT. In particular, CMT + LDR-L1 reaches a final ESS of $82.98\,\%$ in approximately 130 minutes of wall-clock time on an NVIDIA A100 GPU (40 GB memory) using only $10^6$ target energy evaluations. Non-regularized CMT yields very poor performance in this setting.

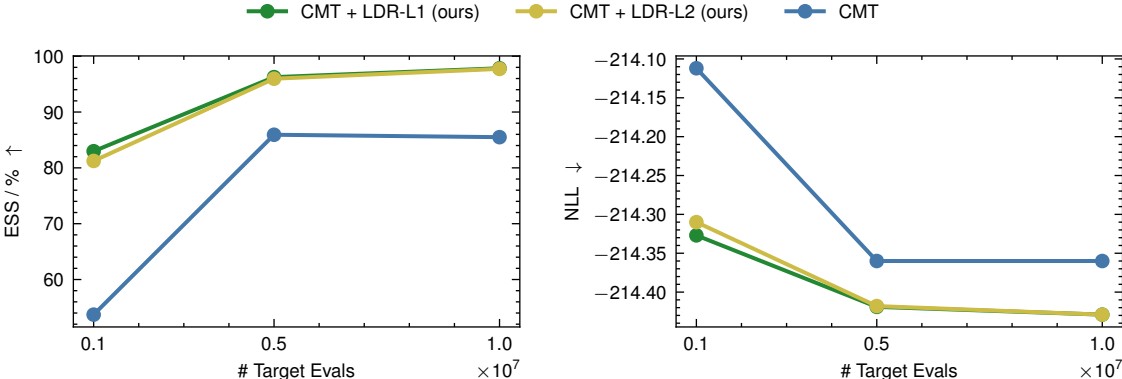

*Figure 8.* Final ESS and NLL for a total of $10^6$, $5 \times 10^6$, and $10^7$ energy evaluations.

### D.5. Connection to the Path-Gradient Forward KL

Originally, path gradients were developed as a method to obtain a lower-variance gradient estimator of the reverse KL objective (Vaitl et al., 2022). Vaitl et al. (2024) noted that the forward KL in data space can be interpreted as a reverse KL in the latent space. This allows one to apply the path-gradient idea also to data-based forward KL training. The objective that results from this approach is a data-based objective that includes target gradient information. Vaitl et al. (2024) hypothesized that the inclusion of target gradients yields regularized training compared to the standard forward KL and showed this empirically.

Both LDR and the path-gradient forward KL can be interpreted as variance-reduction techniques applied at different levels: path gradients reduce gradient estimator variance, while LDR reduces dispersion in the importance weights directly. To achieve this, LDR leverages target energies (zero-order regularizer), while the path-gradient forward KL uses target gradients (first-order regularizer).

LDR is a flexible regularization framework that can be combined with any data-based objective, including the path-gradient forward KL. Combining both yields a doubly regularized objective: path-gradient forward KL provides a first-order regularizer (target gradients), while LDR adds a zero-order regularizer (target energies). Empirical evaluation of this combined approach is an interesting direction for future work.

**Comparison of the regularizing effect**   We directly compare the regularizing effect of forward KL + LDR with the path gradient forward KL for the Gaussian mixture system and on alanine dipeptide using Cartesian coordinates (see Table 1 and Table 14).

We did not observe stable training on the molecular tasks with internal coordinates using neural spline flows. We tested several variants of gradient norm clipping and tuned the learning rate extensively. We hypothesize that the problem lies in the non-continuity of the gradients of the log-density of the neural spline flow (due to the periodic handling of torsion angles), which makes regularization using gradients of the target density difficult.

# E. Extended Results

*Table 10.* Results for training on unbiased datasets from the target distribution of the GMM system. We average over 4 independent experiments with standard deviations. For each number of samples, **bold** indicates metrics not significantly different from the best under uncorrected two-sided Welch's t-tests ($\alpha = 0.05$), capturing seed-level variability only.

| SYSTEM | # SAMPLES | METHOD | NLL ↓ | ESS / % ↑ |
|---|---|---|---|---|
| GMM | 500 | FWD KL | $2.850 \pm 0.061$ | $77.76 \pm 6.90$ |
| | | FWD KL (PATH GRADIENTS) | $2.812 \pm 0.008$ | $77.97 \pm 5.32$ |
| | | FWD KL + LDR-L2 (OURS) | $\mathbf{2.736 \pm 0.003}$ | $95.26 \pm 2.48$ |
| | | FWD KL + LDR-L1 (OURS) | $\mathbf{2.734 \pm 0.004}$ | $\mathbf{96.98 \pm 0.60}$ |
| | 1000 | FWD KL | $2.791 \pm 0.013$ | $86.47 \pm 1.43$ |
| | | FWD KL (PATH GRADIENTS) | $2.777 \pm 0.027$ | $86.82 \pm 5.43$ |
| | | FWD KL + LDR-L2 (OURS) | $\mathbf{2.725 \pm 0.001}$ | $\mathbf{98.62 \pm 0.31}$ |
| | | FWD KL + LDR-L1 (OURS) | $\mathbf{2.727 \pm 0.004}$ | $96.83 \pm 2.60$ |
| | 10000 | FWD KL | $2.733 \pm 0.003$ | $96.75 \pm 0.46$ |
| | | FWD KL (PATH GRADIENTS) | $2.726 \pm 0.002$ | $\mathbf{97.54 \pm 1.44}$ |
| | | FWD KL + LDR-L2 (OURS) | $\mathbf{2.730 \pm 0.014}$ | $\mathbf{97.54 \pm 2.50}$ |
| | | FWD KL + LDR-L1 (OURS) | $\mathbf{2.720 \pm 0.001}$ | $\mathbf{99.47 \pm 0.29}$ |

*Table 11.* Results for training on unbiased datasets from the target distribution of alanine dipeptide and alanine hexapeptide. We average over 4 independent experiments with standard deviations. For each number of samples, **bold** indicates metrics not significantly different from the best under uncorrected two-sided Welch's t-tests ($\alpha = 0.05$), capturing seed-level variability only.

| SYSTEM | # SAMPLES | METHOD | NLL ↓ | ESS / % ↑ | RAM KL ↓ | RAM KL W. RW ↓ | TICA KL ↓ | TICA KL W. RW ↓ | $\mathcal{E}$-$\mathcal{W}_2$ |
|---|---|---|---|---|---|---|---|---|---|
| ALANINE DIPEPTIDE | $1 \times 10^6$ | FWD KL | $-214.378 \pm 0.000$ | $87.94 \pm 0.06$ | $(2.57 \pm 0.12) \times 10^{-3}$ | $(1.96 \pm 0.04) \times 10^{-3}$ | $\mathbf{(1.18 \pm 0.18) \times 10^{-3}}$ | $(1.05 \pm 0.07) \times 10^{-3}$ | $\mathbf{(5.99 \pm 1.47) \times 10^{-3}}$ |
| | | FWD KL + LDR-L2 (OURS) | $\mathbf{-214.432 \pm 0.001}$ | $\mathbf{98.25 \pm 0.11}$ | $\mathbf{(1.60 \pm 0.05) \times 10^{-3}}$ | $\mathbf{(1.34 \pm 0.03) \times 10^{-3}}$ | $(9.58 \pm 0.20) \times 10^{-4}$ | $(7.05 \pm 0.34) \times 10^{-4}$ | $\mathbf{(4.96 \pm 0.64) \times 10^{-3}}$ |
| | | FWD KL + LDR-L1 (OURS) | $-214.430 \pm 0.000$ | $97.81 \pm 0.07$ | $\mathbf{(1.64 \pm 0.04) \times 10^{-3}}$ | $(1.55 \pm 0.03) \times 10^{-3}$ | $(9.71 \pm 0.91) \times 10^{-4}$ | $(8.40 \pm 0.87) \times 10^{-4}$ | $\mathbf{(5.88 \pm 0.51) \times 10^{-3}}$ |
| | $2 \times 10^6$ | FWD KL | $-214.396 \pm 0.001$ | $91.06 \pm 0.22$ | $(1.79 \pm 0.05) \times 10^{-3}$ | $(1.68 \pm 0.06) \times 10^{-3}$ | $\mathbf{(8.74 \pm 0.48) \times 10^{-4}}$ | $(9.21 \pm 0.42) \times 10^{-4}$ | $\mathbf{(6.32 \pm 1.93) \times 10^{-3}}$ |
| | | FWD KL + LDR-L2 (OURS) | $\mathbf{-214.435 \pm 0.001}$ | $\mathbf{98.80 \pm 0.11}$ | $\mathbf{(1.42 \pm 0.04) \times 10^{-3}}$ | $\mathbf{(1.18 \pm 0.04) \times 10^{-3}}$ | $\mathbf{(9.10 \pm 0.94) \times 10^{-4}}$ | $\mathbf{(6.51 \pm 0.31) \times 10^{-4}}$ | $\mathbf{(6.32 \pm 1.17) \times 10^{-3}}$ |
| | | FWD KL + LDR-L1 (OURS) | $\mathbf{-214.435 \pm 0.000}$ | $\mathbf{98.81 \pm 0.04}$ | $\mathbf{(1.48 \pm 0.02) \times 10^{-3}}$ | $\mathbf{(1.36 \pm 0.04) \times 10^{-3}}$ | $\mathbf{(8.62 \pm 0.49) \times 10^{-4}}$ | $(7.18 \pm 0.34) \times 10^{-4}$ | $\mathbf{(6.00 \pm 1.86) \times 10^{-3}}$ |
| | $5 \times 10^6$ | FWD KL | $-214.417 \pm 0.001$ | $94.96 \pm 0.14$ | $(1.49 \pm 0.02) \times 10^{-3}$ | $(1.42 \pm 0.03) \times 10^{-3}$ | $\mathbf{(7.21 \pm 0.20) \times 10^{-4}}$ | $\mathbf{(7.17 \pm 0.32) \times 10^{-4}}$ | $\mathbf{(5.83 \pm 1.54) \times 10^{-3}}$ |
| | | FWD KL + LDR-L2 (OURS) | $\mathbf{-214.437 \pm 0.000}$ | $\mathbf{99.04 \pm 0.02}$ | $\mathbf{(1.34 \pm 0.03) \times 10^{-3}}$ | $\mathbf{(1.28 \pm 0.03) \times 10^{-3}}$ | $(7.87 \pm 0.42) \times 10^{-4}$ | $\mathbf{(6.92 \pm 0.26) \times 10^{-4}}$ | $\mathbf{(5.23 \pm 0.44) \times 10^{-3}}$ |
| | | FWD KL + LDR-L1 (OURS) | $\mathbf{-214.437 \pm 0.001}$ | $\mathbf{99.09 \pm 0.12}$ | $\mathbf{(1.38 \pm 0.04) \times 10^{-3}}$ | $\mathbf{(1.29 \pm 0.04) \times 10^{-3}}$ | $(7.94 \pm 0.13) \times 10^{-4}$ | $\mathbf{(6.91 \pm 0.26) \times 10^{-4}}$ | $\mathbf{(6.37 \pm 1.06) \times 10^{-3}}$ |
| ALANINE HEXAPEPTIDE | $1 \times 10^6$ | FWD KL | $-504.675 \pm 0.002$ | $13.10 \pm 0.11$ | $\mathbf{(3.38 \pm 0.52) \times 10^{-3}}$ | $(6.50 \pm 0.18) \times 10^{-3}$ | $\mathbf{(4.47 \pm 0.60) \times 10^{-3}}$ | $(1.04 \pm 0.06) \times 10^{-2}$ | $\mathbf{(6.93 \pm 0.93) \times 10^{-2}}$ |
| | | FWD KL + LDR-L2 (OURS) | $-504.632 \pm 0.016$ | $14.92 \pm 0.23$ | $(1.29 \pm 0.09) \times 10^{-2}$ | $\mathbf{(4.31 \pm 0.04) \times 10^{-3}}$ | $(2.37 \pm 0.22) \times 10^{-2}$ | $\mathbf{(7.01 \pm 0.30) \times 10^{-3}}$ | $\mathbf{(4.75 \pm 1.19) \times 10^{-2}}$ |
| | | FWD KL + LDR-L1 (OURS) | $\mathbf{-504.759 \pm 0.013}$ | $\mathbf{16.35 \pm 0.10}$ | $(5.04 \pm 0.30) \times 10^{-3}$ | $(4.89 \pm 0.13) \times 10^{-3}$ | $(8.49 \pm 0.40) \times 10^{-3}$ | $(7.86 \pm 0.23) \times 10^{-3}$ | $\mathbf{(5.52 \pm 1.54) \times 10^{-2}}$ |
| | $2 \times 10^6$ | FWD KL | $-504.813 \pm 0.008$ | $16.33 \pm 0.19$ | $(3.06 \pm 0.10) \times 10^{-3}$ | $(5.80 \pm 0.17) \times 10^{-3}$ | $\mathbf{(4.29 \pm 0.23) \times 10^{-3}}$ | $(9.36 \pm 0.26) \times 10^{-3}$ | $\mathbf{(5.67 \pm 0.56) \times 10^{-2}}$ |
| | | FWD KL + LDR-L2 (OURS) | $-505.034 \pm 0.007$ | $25.13 \pm 0.24$ | $(3.66 \pm 0.17) \times 10^{-3}$ | $\mathbf{(3.26 \pm 0.03) \times 10^{-3}}$ | $(5.95 \pm 0.38) \times 10^{-3}$ | $\mathbf{(5.24 \pm 0.09) \times 10^{-3}}$ | $\mathbf{(5.17 \pm 0.69) \times 10^{-2}}$ |
| | | FWD KL + LDR-L1 (OURS) | $\mathbf{-505.066 \pm 0.001}$ | $\mathbf{26.10 \pm 0.12}$ | $\mathbf{(2.84 \pm 0.12) \times 10^{-3}}$ | $(3.35 \pm 0.03) \times 10^{-3}$ | $(4.52 \pm 0.13) \times 10^{-3}$ | $(5.50 \pm 0.06) \times 10^{-3}$ | $\mathbf{(4.71 \pm 0.87) \times 10^{-2}}$ |
| | $5 \times 10^6$ | FWD KL | $-505.174 \pm 0.005$ | $29.60 \pm 0.25$ | $(1.89 \pm 0.04) \times 10^{-3}$ | $(5.39 \pm 0.04) \times 10^{-3}$ | $\mathbf{(2.73 \pm 0.12) \times 10^{-3}}$ | $(5.39 \pm 0.07) \times 10^{-3}$ | $\mathbf{(3.66 \pm 0.63) \times 10^{-2}}$ |
| | | FWD KL + LDR-L2 (OURS) | $-505.259 \pm 0.010$ | $38.70 \pm 0.55$ | $(3.30 \pm 0.38) \times 10^{-3}$ | $\mathbf{(2.44 \pm 0.05) \times 10^{-3}}$ | $(5.64 \pm 0.51) \times 10^{-3}$ | $(4.26 \pm 0.10) \times 10^{-3}$ | $\mathbf{(4.18 \pm 0.28) \times 10^{-2}}$ |
| | | FWD KL + LDR-L1 (OURS) | $\mathbf{-505.308 \pm 0.007}$ | $\mathbf{39.59 \pm 0.44}$ | $\mathbf{(1.73 \pm 0.05) \times 10^{-3}}$ | $(2.63 \pm 0.05) \times 10^{-3}$ | $\mathbf{(2.90 \pm 0.09) \times 10^{-3}}$ | $(4.49 \pm 0.10) \times 10^{-3}$ | $\mathbf{(4.38 \pm 0.73) \times 10^{-2}}$ |

*Table 12.* Results for training on biased datasets using importance sampling. We average over 4 independent experiments with standard deviations. For each number of IS samples, **bold** indicates metrics not significantly different from the best under uncorrected two-sided Welch's t-tests ($\alpha = 0.05$), capturing seed-level variability only.

| SYSTEM | # IS SAMPLES | METHOD | NLL ↓ | ESS / % ↑ | RAM KL ↓ | RAM KL W. RW ↓ | TICA KL ↓ | TICA KL W. RW ↓ | $\mathcal{E}$-$\mathcal{W}_2$ |
|---|---|---|---|---|---|---|---|---|---|
| ALANINE DIPEPTIDE | $1 \times 10^5$ | FWD KL | $-214.114 \pm 0.004$ | $51.42 \pm 0.42$ | $(1.61 \pm 0.07) \times 10^{-2}$ | $(2.21 \pm 0.29) \times 10^{-3}$ | $(4.54 \pm 0.34) \times 10^{-3}$ | $(1.19 \pm 0.11) \times 10^{-3}$ | $(1.25 \pm 0.40) \times 10^{-2}$ |
| | | FWD KL + LDR-L2 ONLY ON IS (OURS) | $-214.264 \pm 0.001$ | $69.04 \pm 0.13$ | $(7.17 \pm 0.69) \times 10^{-3}$ | $(1.49 \pm 0.09) \times 10^{-3}$ | $(2.79 \pm 0.36) \times 10^{-3}$ | $(7.71 \pm 0.32) \times 10^{-4}$ | $(7.53 \pm 0.80) \times 10^{-3}$ |
| | | FWD KL + LDR-L1 ONLY ON IS (OURS) | $-214.251 \pm 0.001$ | $66.94 \pm 0.24$ | $(6.69 \pm 0.15) \times 10^{-3}$ | $(1.62 \pm 0.11) \times 10^{-3}$ | $(2.93 \pm 0.28) \times 10^{-3}$ | $(8.22 \pm 0.60) \times 10^{-4}$ | $\mathbf{(1.01 \pm 0.30) \times 10^{-2}}$ |
| | | FWD KL + LDR-L2 ON BOTH (OURS) | $\mathbf{-214.353 \pm 0.001}$ | $\mathbf{83.20 \pm 0.19}$ | $\mathbf{(2.57 \pm 0.11) \times 10^{-3}}$ | $(1.29 \pm 0.03) \times 10^{-3}$ | $\mathbf{(1.22 \pm 0.06) \times 10^{-3}}$ | $(6.77 \pm 0.44) \times 10^{-4}$ | $\mathbf{(5.76 \pm 0.80) \times 10^{-3}}$ |
| | | FWD KL + LDR-L1 ON BOTH (OURS) | $-214.345 \pm 0.001$ | $82.30 \pm 0.13$ | $(3.26 \pm 0.19) \times 10^{-3}$ | $\mathbf{(1.23 \pm 0.03) \times 10^{-3}}$ | $(1.51 \pm 0.19) \times 10^{-3}$ | $\mathbf{(6.10 \pm 0.13) \times 10^{-4}}$ | $\mathbf{(7.00 \pm 1.94) \times 10^{-3}}$ |
| | $1 \times 10^6$ | FWD KL | $-214.287 \pm 0.001$ | $73.45 \pm 0.15$ | $(5.75 \pm 0.32) \times 10^{-3}$ | $(2.56 \pm 0.24) \times 10^{-3}$ | $(2.30 \pm 0.26) \times 10^{-3}$ | $(1.37 \pm 0.20) \times 10^{-3}$ | $\mathbf{(6.50 \pm 1.46) \times 10^{-3}}$ |
| | | FWD KL + LDR-L2 ONLY ON IS (OURS) | $-214.406 \pm 0.000$ | $92.69 \pm 0.03$ | $(1.88 \pm 0.02) \times 10^{-3}$ | $\mathbf{(1.36 \pm 0.07) \times 10^{-3}}$ | $(1.10 \pm 0.03) \times 10^{-3}$ | $(7.37 \pm 0.31) \times 10^{-4}$ | $\mathbf{(6.26 \pm 1.07) \times 10^{-3}}$ |
| | | FWD KL + LDR-L1 ONLY ON IS (OURS) | $-214.406 \pm 0.000$ | $93.38 \pm 0.03$ | $(1.73 \pm 0.02) \times 10^{-3}$ | $(1.27 \pm 0.04) \times 10^{-3}$ | $(1.04 \pm 0.03) \times 10^{-3}$ | $(6.73 \pm 0.28) \times 10^{-4}$ | $\mathbf{(5.38 \pm 0.66) \times 10^{-3}}$ |
| | | FWD KL + LDR-L2 ON BOTH (OURS) | $\mathbf{-214.416 \pm 0.000}$ | $94.91 \pm 0.05$ | $\mathbf{(1.61 \pm 0.04) \times 10^{-3}}$ | $(1.31 \pm 0.05) \times 10^{-3}$ | $\mathbf{(8.48 \pm 0.71) \times 10^{-4}}$ | $(6.86 \pm 0.65) \times 10^{-4}$ | $\mathbf{(5.49 \pm 1.24) \times 10^{-3}}$ |
| | | FWD KL + LDR-L1 ON BOTH (OURS) | $\mathbf{-214.416 \pm 0.000}$ | $\mathbf{95.28 \pm 0.05}$ | $\mathbf{(1.60 \pm 0.01) \times 10^{-3}}$ | $\mathbf{(1.28 \pm 0.01) \times 10^{-3}}$ | $\mathbf{(8.61 \pm 0.34) \times 10^{-4}}$ | $\mathbf{(6.42 \pm 0.22) \times 10^{-4}}$ | $\mathbf{(6.78 \pm 1.84) \times 10^{-3}}$ |
| | $1 \times 10^7$ | FWD KL | $-214.383 \pm 0.000$ | $88.85 \pm 0.03$ | $(1.91 \pm 0.05) \times 10^{-3}$ | $(1.64 \pm 0.04) \times 10^{-3}$ | $(8.97 \pm 0.36) \times 10^{-4}$ | $(8.91 \pm 0.27) \times 10^{-4}$ | $\mathbf{(6.08 \pm 1.49) \times 10^{-3}}$ |
| | | FWD KL + LDR-L2 ONLY ON IS (OURS) | $-214.436 \pm 0.000$ | $\mathbf{98.78 \pm 0.01}$ | $\mathbf{(1.37 \pm 0.05) \times 10^{-3}}$ | $\mathbf{(1.30 \pm 0.04) \times 10^{-3}}$ | $(8.24 \pm 0.49) \times 10^{-4}$ | $(7.17 \pm 0.48) \times 10^{-4}$ | $\mathbf{(5.29 \pm 1.19) \times 10^{-3}}$ |
| | | FWD KL + LDR-L1 ONLY ON IS (OURS) | $-214.434 \pm 0.000$ | $98.44 \pm 0.01$ | $\mathbf{(1.37 \pm 0.05) \times 10^{-3}}$ | $\mathbf{(1.25 \pm 0.05) \times 10^{-3}}$ | $\mathbf{(8.07 \pm 0.32) \times 10^{-4}}$ | $(6.73 \pm 0.39) \times 10^{-4}$ | $\mathbf{(5.63 \pm 1.53) \times 10^{-3}}$ |
| | | FWD KL + LDR-L2 ON BOTH (OURS) | $-214.434 \pm 0.000$ | $98.48 \pm 0.02$ | $\mathbf{(1.41 \pm 0.05) \times 10^{-3}}$ | $\mathbf{(1.34 \pm 0.05) \times 10^{-3}}$ | $(8.26 \pm 0.52) \times 10^{-4}$ | $\mathbf{(6.99 \pm 0.20) \times 10^{-4}}$ | $\mathbf{(6.31 \pm 2.38) \times 10^{-3}}$ |
| | | FWD KL + LDR-L1 ON BOTH (OURS) | $-214.431 \pm 0.000$ | $97.97 \pm 0.03$ | $(1.45 \pm 0.04) \times 10^{-3}$ | $\mathbf{(1.31 \pm 0.04) \times 10^{-3}}$ | $(8.11 \pm 0.22) \times 10^{-4}$ | $\mathbf{(6.73 \pm 0.21) \times 10^{-4}}$ | $\mathbf{(6.64 \pm 1.43) \times 10^{-3}}$ |

*Table 13.* Results for variational training without data from the target distribution. We average over 4 independent experiments with standard deviations for alanine dipeptide and 3 independent experiments for alanine hexapeptide. For each number of target evaluations, **bold** indicates metrics not significantly different from the best under uncorrected two-sided Welch's t-tests ($\alpha = 0.05$), capturing seed-level variability only.

| SYSTEM | # TARGET EVALS | METHOD | NLL ↓ | ESS / % ↑ | RAM KL ↓ | RAM KL W. RW ↓ | TICA KL ↓ | TICA KL W. RW ↓ | $\mathcal{E}\text{-}\mathcal{W}_2$ |
|---|---|---|---|---|---|---|---|---|---|
| | $2.13 \times 10^8$ | FAB | $-214.412 \pm 0.001$ | $94.95 \pm 0.12$ | $(1.51 \pm 0.04) \times 10^{-3}$ | $(1.22 \pm 0.03) \times 10^{-3}$ | $(3.05 \pm 0.40) \times 10^{-3}$ | $(6.30 \pm 0.12) \times 10^{-4}$ | $(5.89 \pm 0.77) \times 10^{-3}$ |
| | $1 \times 10^8$ | TA-BG | $-214.419 \pm 0.002$ | $95.76 \pm 0.25$ | $(1.91 \pm 0.13) \times 10^{-3}$ | $(1.34 \pm 0.06) \times 10^{-3}$ | $(2.72 \pm 0.80) \times 10^{-3}$ | $(1.08 \pm 0.17) \times 10^{-3}$ | $(8.70 \pm 1.71) \times 10^{-3}$ |
| | $1 \times 10^7$ | CMT | $-214.358 \pm 0.003$ | $85.20 \pm 0.47$ | $(3.78 \pm 0.38) \times 10^{-3}$ | $(2.33 \pm 0.23) \times 10^{-3}$ | $(2.16 \pm 0.71) \times 10^{-3}$ | $(1.17 \pm 0.12) \times 10^{-3}$ | $\mathbf{(5.36 \pm 1.20) \times 10^{-3}}$ |
| | | CMT + LDR-L2 (OURS) | $\mathbf{-214.429 \pm 0.001}$ | $\mathbf{97.81 \pm 0.08}$ | $\mathbf{(1.62 \pm 0.05) \times 10^{-3}}$ | $\mathbf{(1.41 \pm 0.04) \times 10^{-3}}$ | $\mathbf{(9.36 \pm 0.07) \times 10^{-4}}$ | $\mathbf{(7.43 \pm 0.40) \times 10^{-4}}$ | $\mathbf{(5.66 \pm 0.26) \times 10^{-3}}$ |
| | | CMT + LDR-L1 (OURS) | $\mathbf{-214.429 \pm 0.001}$ | $\mathbf{97.84 \pm 0.05}$ | $\mathbf{(1.57 \pm 0.09) \times 10^{-3}}$ | $\mathbf{(1.42 \pm 0.08) \times 10^{-3}}$ | $\mathbf{(9.47 \pm 0.44) \times 10^{-4}}$ | $\mathbf{(7.46 \pm 0.70) \times 10^{-4}}$ | $\mathbf{(5.70 \pm 0.65) \times 10^{-3}}$ |
| ALANINE DIPEPTIDE | $2 \times 10^7$ | CMT | $-214.405 \pm 0.001$ | $93.18 \pm 0.19$ | $(1.70 \pm 0.03) \times 10^{-3}$ | $(1.52 \pm 0.03) \times 10^{-3}$ | $\mathbf{(1.40 \pm 0.33) \times 10^{-3}}$ | $\mathbf{(7.94 \pm 0.68) \times 10^{-4}}$ | $\mathbf{(4.84 \pm 0.90) \times 10^{-3}}$ |
| | | CMT + LDR-L2 (OURS) | $-214.432 \pm 0.000$ | $98.41 \pm 0.03$ | $\mathbf{(1.38 \pm 0.03) \times 10^{-3}}$ | $\mathbf{(1.29 \pm 0.06) \times 10^{-3}}$ | $(1.95 \pm 0.10) \times 10^{-3}$ | $(9.73 \pm 0.55) \times 10^{-4}$ | $\mathbf{(6.08 \pm 1.21) \times 10^{-3}}$ |
| | | CMT + LDR-L1 (OURS) | $\mathbf{-214.435 \pm 0.001}$ | $\mathbf{98.81 \pm 0.09}$ | $\mathbf{(1.47 \pm 0.10) \times 10^{-3}}$ | $\mathbf{(1.29 \pm 0.10) \times 10^{-3}}$ | $\mathbf{(1.56 \pm 0.47) \times 10^{-3}}$ | $\mathbf{(8.96 \pm 1.75) \times 10^{-4}}$ | $(8.37 \pm 1.48) \times 10^{-3}$ |
| | $1 \times 10^8$ | CMT | $-214.431 \pm 0.000$ | $97.85 \pm 0.05$ | $\mathbf{(1.46 \pm 0.04) \times 10^{-3}}$ | $(1.39 \pm 0.03) \times 10^{-3}$ | $\mathbf{(1.78 \pm 0.04) \times 10^{-3}}$ | $\mathbf{(1.09 \pm 0.04) \times 10^{-3}}$ | $(6.75 \pm 2.05) \times 10^{-3}$ |
| | | CMT + LDR-L2 (OURS) | $\mathbf{-214.437 \pm 0.001}$ | $99.11 \pm 0.06$ | $\mathbf{(1.39 \pm 0.04) \times 10^{-3}}$ | $\mathbf{(1.36 \pm 0.03) \times 10^{-3}}$ | $(1.94 \pm 0.42) \times 10^{-3}$ | $\mathbf{(1.11 \pm 0.08) \times 10^{-3}}$ | $(7.65 \pm 2.06) \times 10^{-3}$ |
| | | CMT + LDR-L1 (OURS) | $\mathbf{-214.438 \pm 0.000}$ | $\mathbf{99.29 \pm 0.04}$ | $\mathbf{(1.42 \pm 0.03) \times 10^{-3}}$ | $\mathbf{(1.32 \pm 0.02) \times 10^{-3}}$ | $\mathbf{(1.81 \pm 0.05) \times 10^{-3}}$ | $\mathbf{(1.12 \pm 0.07) \times 10^{-3}}$ | $(8.92 \pm 0.64) \times 10^{-3}$ |
| | $4.2 \times 10^8$ | FAB | $-504.355 \pm 0.019$ | $14.41 \pm 0.27$ | $(2.15 \pm 0.12) \times 10^{-2}$ | $(1.02 \pm 0.05) \times 10^{-2}$ | $(3.62 \pm 0.06) \times 10^{-2}$ | $(1.57 \pm 0.07) \times 10^{-2}$ | $(3.38 \pm 1.25) \times 10^{-2}$ |
| | $4 \times 10^8$ | TA-BG | $-504.782 \pm 0.019$ | $18.66 \pm 0.16$ | $(6.74 \pm 1.29) \times 10^{-3}$ | $(6.74 \pm 0.89) \times 10^{-3}$ | $(8.19 \pm 1.17) \times 10^{-3}$ | $(1.14 \pm 0.10) \times 10^{-2}$ | $(1.60 \pm 0.49) \times 10^{-2}$ |
| | $1 \times 10^8$ | CMT | $-503.190 \pm 0.051$ | $5.92 \pm 0.37$ | $(6.72 \pm 0.73) \times 10^{-2}$ | $(5.23 \pm 0.51) \times 10^{-2}$ | $(1.56 \pm 0.45) \times 10^{-1}$ | $\mathbf{(1.07 \pm 0.43) \times 10^{-1}}$ | $(8.30 \pm 1.78) \times 10^{-2}$ |
| | | CMT + LDR-L2 (OURS) | $\mathbf{-504.801 \pm 0.008}$ | $31.47 \pm 0.71$ | $\mathbf{(1.61 \pm 0.08) \times 10^{-2}}$ | $\mathbf{(1.56 \pm 0.23) \times 10^{-2}}$ | $\mathbf{(1.33 \pm 0.06) \times 10^{-2}}$ | $\mathbf{(1.58 \pm 0.08) \times 10^{-2}}$ | $\mathbf{(5.92 \pm 1.28) \times 10^{-2}}$ |
| | | CMT + LDR-L1 (OURS) | $-504.721 \pm 0.010$ | $29.48 \pm 0.40$ | $(1.71 \pm 0.03) \times 10^{-2}$ | $(1.65 \pm 0.05) \times 10^{-2}$ | $\mathbf{(1.30 \pm 0.15) \times 10^{-2}}$ | $\mathbf{(1.77 \pm 0.14) \times 10^{-2}}$ | $(6.35 \pm 0.33) \times 10^{-2}$ |
| ALANINE HEXAPEPTIDE | $2 \times 10^8$ | CMT | $-503.545 \pm 0.364$ | $20.85 \pm 0.65$ | $(9.08 \pm 3.16) \times 10^{-2}$ | $\mathbf{(8.32 \pm 3.59) \times 10^{-2}}$ | $(1.89 \pm 0.19) \times 10^{-1}$ | $(1.41 \pm 0.24) \times 10^{-1}$ | $\mathbf{(3.96 \pm 1.52) \times 10^{-2}}$ |
| | | CMT + LDR-L2 (OURS) | $-505.012 \pm 0.004$ | $41.71 \pm 0.47$ | $(9.05 \pm 0.65) \times 10^{-3}$ | $(1.02 \pm 0.03) \times 10^{-2}$ | $\mathbf{(8.81 \pm 0.50) \times 10^{-3}}$ | $(1.12 \pm 0.01) \times 10^{-2}$ | $\mathbf{(3.32 \pm 0.54) \times 10^{-2}}$ |
| | | CMT + LDR-L1 (OURS) | $\mathbf{-505.051 \pm 0.013}$ | $43.66 \pm 0.69$ | $\mathbf{(7.78 \pm 0.06) \times 10^{-3}}$ | $\mathbf{(8.30 \pm 0.28) \times 10^{-3}}$ | $\mathbf{(8.65 \pm 0.28) \times 10^{-3}}$ | $\mathbf{(1.07 \pm 0.04) \times 10^{-2}}$ | $\mathbf{(2.96 \pm 0.64) \times 10^{-2}}$ |
| | $4 \times 10^8$ | CMT | $-504.770 \pm 0.037$ | $23.95 \pm 0.34$ | $(1.39 \pm 0.07) \times 10^{-2}$ | $(1.27 \pm 0.06) \times 10^{-2}$ | $(1.54 \pm 0.08) \times 10^{-2}$ | $(1.88 \pm 0.05) \times 10^{-2}$ | $(6.17 \pm 0.55) \times 10^{-2}$ |
| | | CMT + LDR-L2 (OURS) | $\mathbf{-505.193 \pm 0.007}$ | $47.76 \pm 0.31$ | $\mathbf{(5.08 \pm 0.12) \times 10^{-3}}$ | $\mathbf{(5.52 \pm 0.06) \times 10^{-3}}$ | $\mathbf{(6.53 \pm 0.07) \times 10^{-3}}$ | $\mathbf{(7.07 \pm 0.10) \times 10^{-3}}$ | $\mathbf{(1.22 \pm 0.19) \times 10^{-2}}$ |
| | | CMT + LDR-L1 (OURS) | $\mathbf{-505.219 \pm 0.019}$ | $\mathbf{48.32 \pm 0.66}$ | $\mathbf{(5.06 \pm 0.94) \times 10^{-3}}$ | $\mathbf{(5.88 \pm 0.70) \times 10^{-3}}$ | $\mathbf{(6.25 \pm 0.58) \times 10^{-3}}$ | $\mathbf{(7.82 \pm 0.72) \times 10^{-3}}$ | $\mathbf{(1.48 \pm 0.29) \times 10^{-2}}$ |

*Table 14.* Results for training on Cartesian coordinates using an unbiased dataset from the target distribution of alanine dipeptide. We average over 4 independent experiments with standard deviations. **Bold** indicates metrics not significantly different from the best under uncorrected two-sided Welch's t-tests ($\alpha = 0.05$), capturing seed-level variability only.

| METHOD | NLL ↓ | ESS / % ↑ | RAM KL ↓ | RAM KL W. RW ↓ | $\mathcal{E}\text{-}\mathcal{W}_2$ |
|---|---|---|---|---|---|
| FWD KL (Tan et al., 2025a) | $-219.583 \pm 0.051$ | $19.40 \pm 0.80$ | $(2.93 \pm 0.29) \times 10^{-2}$ | $(2.87 \pm 0.06) \times 10^{-2}$ | $(1.51 \pm 0.20) \times 10^{-1}$ |
| FWD KL★ | $-219.583 \pm 0.050$ | $27.32 \pm 1.28$ | $(2.93 \pm 0.30) \times 10^{-2}$ | $(2.51 \pm 0.08) \times 10^{-2}$ | $(1.43 \pm 0.09) \times 10^{-1}$ |
| FWD KL★ (PATH) | $-218.528 \pm 0.090$ | $10.31 \pm 0.86$ | $(1.04 \pm 0.11) \times 10^{-1}$ | $(4.77 \pm 0.95) \times 10^{-2}$ | $(1.77 \pm 0.15) \times 10^{-1}$ |
| FWD KL★ + LDR-L2 (OURS) | $\mathbf{-219.815 \pm 0.034}$ | $\mathbf{35.89 \pm 1.69}$ | $(2.50 \pm 0.35) \times 10^{-2}$ | $\mathbf{(1.79 \pm 0.10) \times 10^{-2}}$ | $\mathbf{(9.27 \pm 2.20) \times 10^{-2}}$ |
| FWD KL★ + LDR-L1 (OURS) | $\mathbf{-219.800 \pm 0.044}$ | $\mathbf{35.75 \pm 1.63}$ | $\mathbf{(2.16 \pm 0.08) \times 10^{-2}}$ | $\mathbf{(2.07 \pm 0.23) \times 10^{-2}}$ | $(1.15 \pm 0.33) \times 10^{-1}$ |

★ Uses our improved augmentation correction, see Appendix C.1.

# F. Visualizations

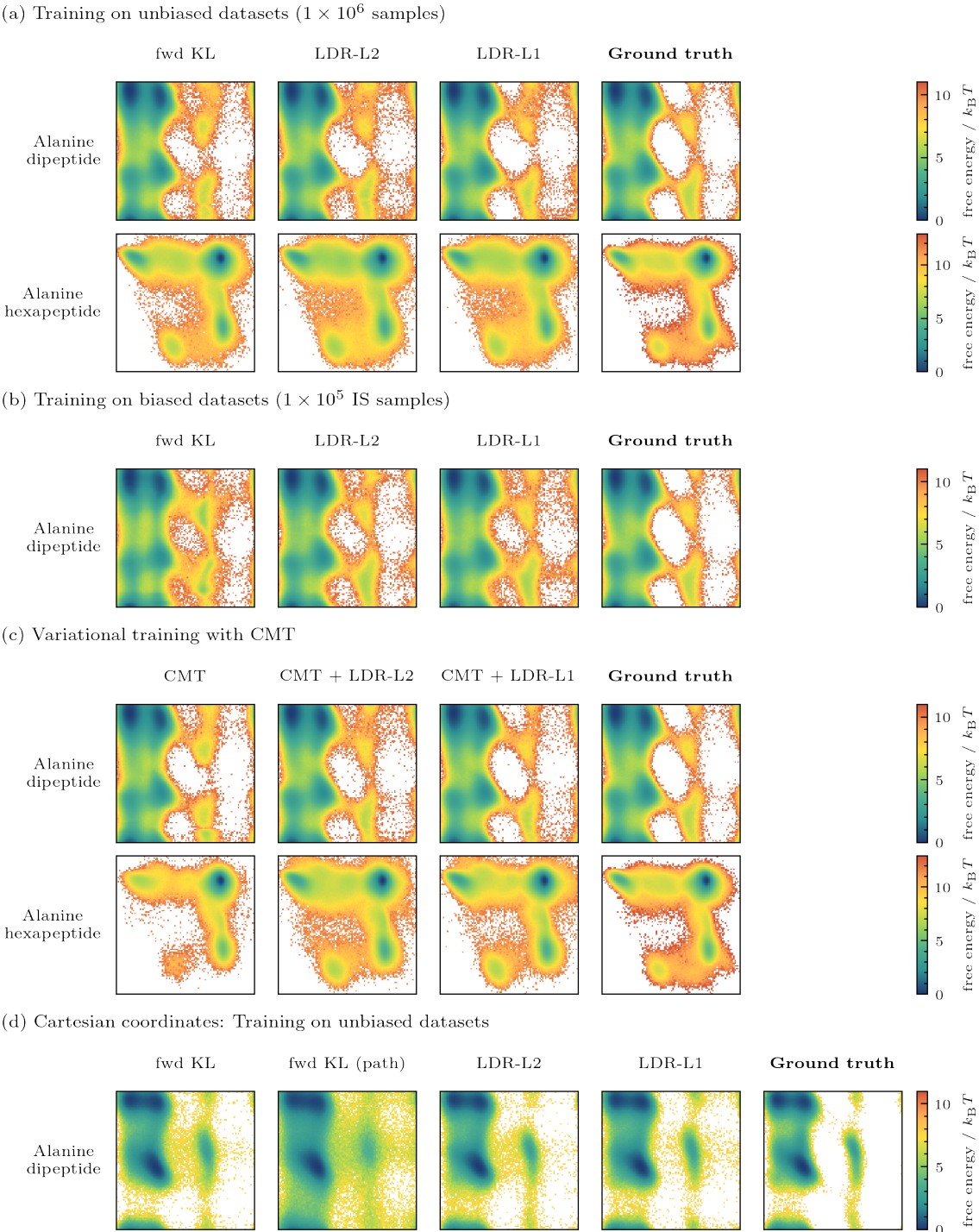

*Figure 9.* Visualization of the 2D marginals of the main degrees of freedom for each system. For alanine dipeptide, we show the marginal of the two main dihedral angles (Ramachandran), for alanine hexapeptide the 2D TICA projection. (a) Training on unbiased datasets using $1 \times 10^6$ samples, (b) training on biased dataset using $1 \times 10^5$ IS samples, (c) variational training with CMT ($1 \times 10^7$ target evaluations for alanine dipeptide, $1 \times 10^8$ for alanine hexapeptide), (d) training in Cartesian coordinates on unbiased dataset ($1 \times 10^5$ samples).

