# OpenReview forum: "Efficient Training of Boltzmann Generators Using Off-Policy Log-Dispersion Regularization"
_ICML.cc/2026/Conference — ICML 2026 regular_

### Official Review · Reviewer_caTY · 2026-03-03

**Soundness:** 3
**Presentation:** 4
**Significance:** 3
**Originality:** 3
**Overall Recommendation:** 5
**Confidence:** 5

**Summary:**

This paper introduces an off-policy regularization method called "Log-Dispersion Regularization" (LDR) to improve the training of Boltzmann Generators (BGs). Traditionally, BGs are trained using forward KL divergence. While reverse KL is sometimes added, it often leads to mode-collapse and introduces computational overhead due to its on-policy nature. To address this, the authors adapt the concept of minimizing the log-variance of importance weights—a technique well-explored in the neural-sampler community—and generalize it into LDR. The paper evaluates both $L=1$ and $L=2$ variants of LDR across a wide range of settings, demonstrating that this off-policy approach improves BG training without adding computational overhead to the energy function evaluation.

**Compliance With Llm Reviewing Policy:**

Affirmed.

**Final Justification:**

The author has addressed my questions, clarified the relation between the path-gradient paper and the mentioned importance sampling method. Overall, I will maintain my score as a clear accept.

**Key Questions For Authors:**

**1. Connection to Path-Gradient Methods:**
* **Question:** Could you elaborate more comprehensively on the theoretical or empirical connections between your proposed LDR method and existing path-gradient method?
* **How it impacts my evaluation:** A clearer explanation of this connection (perhaps added to the appendix or related work) would strengthen the "Presentation" and "Originality" dimensions of my review by better grounding the theory in the broader literature.

**2. Regarding the reference distribution**
* **Question:** As claimed in the paper, the difference between the reference distribution and the target distribution can lead to potential instability during training. To avoid recomputing the energy while reducing this gap, one could do importance sampling to reweight the reference closer to $q_\theta$. Since both $\tilde p$ and $q_\theta$ are available, the IS-reweighting is simple to apply, where the importance weights can be detached as the off-policy nature of the loss. Therefore, perhaps the author could try doing this and see if it gains.
* **How it impacts my evaluation:** This would potentially reduce the gap between the reference and $q_\theta$, and might solve the potential limitation of training instability. However, the reviewer understands that the IS reweighing can be cursed by dimensionality, therefore it is normal that it wouldn't work that well in high-dim. Therefore, even if the author fails to improve the results with this IS-reweighting, the reviewer wouldn't decrease the score.

**Limitations:**

yes

**Strengths And Weaknesses:**

# Soundness

The proposed LDR method is technically sound and backed by a robust set of experiments across diverse settings. The empirical results are promising and clearly demonstrate that the method successfully avoids the mode-collapse issues associated with reverse KL, while also bypassing the heavy on-policy computational burdens.

# Presentation
* **Strengths:** The paper is clearly written, well-structured, and the overall narrative is easy to follow. The progression from the baseline problem (forward/reverse KL limitations) to the proposed generalized solution (LDR) is logical.
* **Weaknesses:** The connection between the proposed method and path-gradient methods is briefly mentioned but underexplored. Expanding on this relationship would provide a more comprehensive theoretical context for the reader.

# Significance
The paper addresses a highly relevant problem in the generative modeling of physical systems. Because the LDR method improves training without requiring additional evaluations of the energy function—which is typically the main bottleneck in BGs—the practical utility for practitioners is high. It represents a meaningful advancement in how these models can be trained efficiently.

# Originality
The core idea is highly novel and represents a creative bridge between distinct research areas. By successfully migrating and generalizing the minimization of log-variance of importance weights from the neural-sampler community into the context of Boltzmann Generators, the authors have provided a fresh, effective perspective on off-policy regularization.

---

> ### Author Rebuttal · Authors · 2026-03-30
>
> We thank the reviewer for the positive assessment and helpful comments. We address all questions and comments below.
>
> **Connection to path-gradient forward KL**
> We acknowledge that the connection to path gradients was not discussed in appropriate detail in our original manuscript. We will include an extended discussion in the final version.
>
> Originally, path gradients were developed as a method to obtain a lower-variance gradient estimator of the reverse KL objective. Vaitl et al. (2024) noted that the forward KL in data space can be interpreted as a reverse KL in the latent space. This allows one to apply the path-gradient idea also to data-based forward KL training. The objective that results from this approach is a data-based objective that includes target gradient information. Vaitl et al. (2024) hypothesized that the inclusion of target gradients yields regularized training compared to the standard forward KL and showed this empirically.
>
> In contrast, our log-dispersion objective provides a direct intuition for its regularizing effect. Minimizing log-dispersion reduces variability in the log importance weights, which can be interpreted as encouraging the learned energy landscape to match the target up to a global shift, i.e., acting as a shape regularizer. To summarize, both LDR and path gradients can be interpreted as variance-reduction techniques applied at different levels: path gradients reduce gradient estimator variance, while LDR reduces dispersion in the importance weights directly.
>
> We provided an empirical comparison of the regularizing effect of LDR and path gradients in Table 13 in the appendix. Given the same compute budget, LDR-regularized forward KL outperforms the path-gradient-regularized forward KL by a significant margin. We further note that LDR is faster to evaluate and more flexible, as it can be applied to arbitrary reference distributions, not only to the target itself.
>
> Finally, we’d like to mention that we do not actually view LDR as a “competitor” to path-gradient forward KL. LDR is a flexible regularization framework that can be combined with any data-based objective, including the path-gradient forward KL. Combining both yields a doubly regularized objective: path-gradient forward KL provides a first-order regularizer (target gradients), while LDR adds a zero-order regularizer (target energies). We are currently working to include experiments on this doubly-regularized approach in the final version.
>
> **Avoiding instabilities**
> We noted in our manuscript that instabilities might occur if the (biased) reference distribution is too far away from the target. We thank the reviewer for the suggestion to use importance sampling to reweight the reference distribution closer to the current $q_\theta$.
>
> We agree that this is a natural and interesting extension. However, we note that this is only possible if the density of the biased reference distribution $p_X^{\text {biased }}$ is known. For example, in our biased experiments on alanine dipeptide, the density of the biased distribution is unknown because it arises from a mixture of multiple MD trajectories starting from different minima. One could instead use the density $q_X^{\theta_1}$ from the Boltzmann generator initially trained on the biased distribution. However, this would introduce additional noise, on top of the noise from importance sampling, since the density estimate is only a rough estimate of the underlying biased distribution.
>
> We will mention the option of reweighting the reference distribution for cases where the density of the reference distribution is known, and again thank the reviewer for this suggestion.
>
> **Summary**
> We thank the reviewer for the valuable comments and positive assessment of our work. We have addressed all questions and concerns raised and hope that our revisions further support its strong contribution to the field.

---

> > ### Author Rebuttal · Reviewer_caTY · 2026-04-01
> >
> > Thank you for clarifying the relation between the path-gradient paper and the mentioned importance sampling method. I will maintain my score.

---

> > > ### Author Response · Authors · 2026-04-06
> > >
> > > We are happy that our additional discussion on the connection to path gradients was helpful. We agree that expanding on this will strengthen our manuscript and thank the reviewer for this suggestion.
> > >
> > > Finally, we thank the reviewer for their time and thoughtful engagement with our work during the rebuttal phase. We appreciate the reviewer’s recognition of the practical impact of LDR and hope our framework serves as a solid foundation for future works in this area.

---

### Official Review · Reviewer_WPoV · 2026-03-11

**Soundness:** 3
**Presentation:** 3
**Significance:** 3
**Originality:** 3
**Overall Recommendation:** 4
**Confidence:** 4

**Summary:**

This paper proposes off-policy log-dispersion regularization (LDR), which generalizes the log-variance objective to a family of dispersion measures (p≥1) and applies it as an auxiliary regularizer on top of standard data-based objectives for training Boltzmann generators. LDR leverages target energy labels already available from dataset construction without requiring additional on-policy samples or energy evaluations. Empirically, LDR consistently improves both final performance and data efficiency across three training regimes: unbiased datasets, biased simulation datasets, and purely data-free setting. In the data-free setting, applying LDR to the state-of-the-art Constrained Mass Transport (CMT) method yields up to 10× improvement in target evaluation efficiency on alanine dipeptide.

**Compliance With Llm Reviewing Policy:**

Affirmed.

**Final Justification:**

The authors' rebuttal thoroughly addressed my concerns, including additional ablations, new baselines, and empirical evidence I had requested. I maintain my score of 4 (weak accept), which was already leaning toward acceptance. The paper presents a simple yet effective regularization strategy with comprehensive experimental coverage, and the revisions promised for the camera-ready version should further strengthen the contribution.

**Key Questions For Authors:**

**Q1**. In Section 4.3 (training on biased datasets), a natural baseline would be forward KL pre-training on the biased dataset followed by short fine-tuning with forward KL + reverse KL, allocating the same energy evaluation budget (~1.1×10⁶) to reverse KL training instead of constructing the IS dataset. Since both approaches leverage target energy information, this comparison would help isolate the specific benefit of LDR over more standard energy-based training. Could you clarify why this setting was omitted?


**Q2**. Except for unbiased training on alanine hexapeptide, LDR-L1 and LDR-L2 show very similar performance across all settings. Could you provide empirical evidence (e.g., importance weight distributions, gradient norms, or training loss trajectories) that LDR-L1 offers better training stability or numerical properties, as suggested in Section 5?


**Q3**. In line 383, vanilla CMT on alanine hexapeptide at 1×10⁸ evaluations is described as exhibiting "mode collapse" with an ESS of 5.92%. However, since the reported ESS is the reverse ESS (computed from flow samples), it reflects the concentration of importance weights rather than directly diagnosing missing modes. Could you provide qualitative evidence, such as Ramachandran or TICA plots, confirming that mode collapse actually occurs in this setting?


**Q4**. Since LDR-L2 — which is essentially the classical log-variance objective used as a regularizer — performs comparably to LDR-L1 across nearly all settings, it appears that the key contribution lies in the off-policy regularization strategy itself rather than the generalization from log-variance to the log-dispersion family. More explicitly highlighting this distinction would help readers identify the core contribution of the paper more clearly.

**Limitations:**

Yes

**Strengths And Weaknesses:**

### Strengths
- **S1. Comprehensive experimental coverage.** LDR is evaluated across three distinct training regimes — unbiased datasets, biased simulation datasets, and purely variational training without target samples — as well as two coordinate representations (internal and Cartesian). This breadth convincingly demonstrates that the method is not tied to a specific setting or architecture choice.

- **S2. Simplicity and clear empirical gains.** The method amounts to adding a single regularization term (Equation 5) on top of existing objectives, requires no additional on-policy samples or energy evaluations, and introduces no extra training cost (Table 8). Despite this simplicity, LDR yields consistent and statistically significant improvements across all benchmarks, with data efficiency gains of up to 10× in the variational setting.

### Weaknesses

- **W1. Lack of evidence for some empirical claims.**
In Section 5, the slight advantage of LDR-L1 over LDR-L2 on alanine hexapeptide is attributed to heavy-tailed importance weights, but no direct evidence supports this mechanistic explanation. Importance weight histograms, gradient norm statistics, or training loss curves would substantiate the claim; as it stands, the reader can only observe marginally better final numbers in Table 1 without verifying the proposed mechanism. Similarly, in Section 4.4, LDR is claimed to enable "stable training" for CMT on alanine hexapeptide at 1×10⁸ target evaluations, yet only final metrics are reported. Training curves showing ESS/NLL trajectories over the course of optimization would be needed to properly characterize stability, as opposed to simply better final performance.

- **W2. Insufficient ablations on LDR components.**
Only p=1 and p=2 are tested, despite the generalization to the log-dispersion family (p≥1) being a stated contribution. Given the qualitative difference identified in Appendix A.4 between p=1 (non-vanishing gradients at the optimum) and p≥2 (vanishing gradients), intermediate or higher values (e.g., p=1.5, p=3) would help clarify whether the choice of p matters in practice.
Additionally, the failure mode of LDR-only training (without a data-based objective) is demonstrated only with LDR-L1 (Appendix D.2). Since LDR-L2 penalizes squared deviations more aggressively, it may behave differently in capturing probability mass outside well-sampled regions. Including LDR-L2-only training results would give a more complete picture.

### Minor issues
- The biased dataset construction (Section 4.3, Appendix B.4) relies on prior knowledge of the system's modes, as MD trajectories are initialized from four known minima. Starting from a single point would likely produce a more heavily biased dataset with missing modes, making the comparison of energy evaluations between biased data collection (~10⁶) and unbiased MD simulation (~10⁸) somewhat misleading, since the two approaches assume different levels of prior knowledge.

- (Line 374) The claim that CMT + LDR with 10⁷ evaluations matches CMT with 10⁸ evaluations is based on the ESS-vs-evaluations figure embedded in Table 3, not the table itself. Since CMT's 10⁸ result does not appear in the table body (it is only in Appendix Table 12), explicitly referencing the figure or appendix would improve clarity.

---

> ### Author Rebuttal · Authors · 2026-03-30
>
> We thank the reviewer for the assessment and helpful comments. We address all questions and comments below.
>
> **Further variants of log-dispersion (W2)**
> We appreciate the suggestions on additional log-dispersion variants. In the final version, we will further generalize the log-dispersion objective to
>
> $\mathcal{L}\_{\mathrm{LD}}^\theta=\mathbb{E}\_{r\_X}\left[\rho\left(f^\theta(x)-c^\theta\right)\right], \quad c^\theta=\mathbb{E}\_{r\_X}\left[f^\theta(x)\right],$
>
> where $\rho(u) \geq 0$ and $\rho(u)=0 \Longleftrightarrow u=0$. This recovers LDR-Lp objectives ($\rho(u)=|u|^p$), but now also includes Huber, Student-$t$, and other dispersion penalties.
>
> We performed additional experiments on alanine hexapeptide with p=1.5, p=3, and the Huber objective: https://ibb.co/FkHJjspG. Huber and p=1.5 yield slightly improved results, while p=3 is unstable and performs worse.
>
> We now also show results using only LDR-L1 and only LDR-L2: https://ibb.co/DH8hWyj9. Both fail similarly, indicating that combining LDR with a divergence such as the forward KL is crucial.
>
> **Additional empirical evidence (W1, Q2)**
> We agree with the reviewer on the missing empirical evidence for our claims of improved training stability with LDR-L1 over LDR-L2. The following figure shows the variance of gradient norms within a sliding window during training for LDR-L1, LDR-L2, and the newly added Huber, p=1.5, and p=3: https://ibb.co/fYHdBwwr.
> LDR-L2 exhibits higher variability in gradient norms than LDR-L1, indicating less stable optimization. Additionally, p=3 shows large spikes in gradient norms, explaining its poor performance. We will include these results and expand our discussion on training stability, incorporating gradient norms to support our claims.
>
> **Training stability of CMT (W1, Q3)**
> When stating that LDR enables stable training with CMT on alanine hexapeptide with only $10^8$ target evals, “stable” was intended to imply the absence of mode collapse. We will clarify this.
>
> We agree that evidence beyond final metrics is needed to support the claim that standard CMT undergoes mode collapse, whereas the LDR-regularized variants do not. We thus visualize the TICA projections: https://ibb.co/fzd9fvrn. Mode collapse is clearly visible for standard CMT in the bottom-left. We will highlight this more clearly in the final version of our manuscript.
>
> **Reverse KL baseline when training on biased data (Q1)**
> We agree that reverse KL fine-tuning after pre-training on the biased data is an appropriate additional baseline. We thus show results for fine-tuning with the reverse KL (with a budget of $10^7$ target evaluations): https://ibb.co/KzzZC9pk. Due to the mode-seeking nature of the reverse KL, the proposal distribution undergoes mode collapse. We will include this as an additional baseline.
>
> We note that the reviewer’s suggestion to allocate “the same energy evaluation budget (~1.1×10⁶) to reverse KL training instead of constructing the IS dataset” does not allow a combination of reverse KL and forward KL, since without constructing the IS dataset, forward KL on the target distribution can not be used.
>
> **Biased experiments**
> We agree that our biased experiments assume prior knowledge, which the unbiased experiments do not. Thus, comparing target evaluations can be misleading. We will clarify this in the revision.
>
> We note that the importance sampling approach shown for alanine dipeptide is not the only way to use biased datasets. If a small number of equilibrium samples is given, one can add arbitrary biased datasets to be additionally included in the LDR objective. In this case, the biased distribution need not cover all modes, as no importance sampling is required. Thus, short trajectories with missing modes can still be used without prior knowledge. We are currently working to include such an experiment for alanine hexapeptide in the final version.
>
> **Better highlighting of our contribution (Q4)**
> We agree with the reviewer that using log-dispersion objectives off-policy in combination with data-based divergences, rather than on-policy, is the main contribution of our work. We will highlight this more clearly.
>
> That said, we believe the extension to general dispersion measures is a meaningful contribution. LDR-L1 yields improved performance and more stable optimization in the more challenging experimental setups. Furthermore, the flexibility of our framework suggests that exploring additional penalties could yield valuable additional progress in future work.
>
> **Summary**
> We now further generalized the log-dispersion objective and ran additional experiments, showed previously missing additional empirical evidence for some of our claims, and added an additional reverse KL baseline in the biased setup.
>
> We thank the reviewer for the valuable comments and assessment of our work. We have addressed all questions and concerns raised and hope that our revisions further support its strong contribution to the field.

---

> > ### Author Rebuttal · Reviewer_WPoV · 2026-04-03
> >
> > Thank you for your reply. I will maintain my score, which is already leaning toward acceptance.

---

> > > ### Author Response · Authors · 2026-04-06
> > >
> > > We thank the reviewer for their time and detailed comments during the rebuttal phase. We are happy that we were able to address all comments and questions to the reviewer's satisfaction, and that no open concerns appear to remain.
> > >
> > > In particular, we have:
> > > - further generalized the log-dispersion objective and performed additional ablations,
> > > - provided new empirical evidence supporting our claims on training stability,
> > > - included a new baseline (reverse KL fine-tuning),
> > > - and added TICA visualization to show that LDR avoids mode collapse.
> > >
> > > We will incorporate all of these improvements in the final version to further strengthen the paper. We appreciate the positive assessment of our contribution.

---

### Official Review · Reviewer_JwMY · 2026-03-12

**Soundness:** 3
**Presentation:** 2
**Significance:** 2
**Originality:** 2
**Overall Recommendation:** 4
**Confidence:** 2

**Summary:**

This paper proposes to learn Boltzmann generators by adapting the log-variance objective of Richter et al. (2020) by simply considering other moments of the dispersion (i.e. the energy difference between the model and target distribution) and
combine this log-dispersion objectives with standard data based objectives. An additional distinction concerns whether these dispersion moments are evaluated from the model distribution (on-policy) or a fixed reference distribution (off-policy).
This paper is mainly an  experimental paper.
3 different experimental settings are considered: one using unbiased samples which is the most costly as it requires long molecular dynamics simulations, one using biased samples and one without  samples.

**Compliance With Llm Reviewing Policy:**

Affirmed.

**Final Justification:**

Clarifications provided by the authors have help me to appreciate their contribution, I have raised my score

**Key Questions For Authors:**

- I am not sure to understand the 3rd scenario (without samples) of section 4.4. How are chosen the points corresponding to the target energy evaluations, if not sampled from the target distribution?
are they sampled from the fixed reference distribution r_x? Is the loss {\mathcal L}_D still present in this case?

- I did not find indicated  which reference distribution r_x was chosen.

- I see that p=1 or 2 in the experiments, but I thought that p=2 would correspond to the default model of Richter et al?

- the main improvement of the method seems to be read of from the ESS metric (which is the ratio of the squared average of w(x) over the average of w(x)^2)  which is quite obscure to me (it clearly depend on how the points are sampled, for instance drop mode can affect a lot this metric when equilibrium samples of p_x are considered and not when sampled from q_x)

**Limitations:**

yes

**Strengths And Weaknesses:**

Strong points:
- a good overview of the domain in the introduction (which I admit I am quite ignorant about)
- Empirical evidence indicating that the proposed combined approach improve the results of baseline approaches.


Weak points:

I did not find the paper clearly written,  there are many points I did not get (maybe also because I am not working on this topic).
It seems to me that many important details are missing in the plain text. See questions below.

---

> ### Author Rebuttal · Authors · 2026-03-30
>
> We thank the reviewer for their assessment of our work. We address all comments and questions below.
>
> **Summary of experimental setups**
> While we aimed for clarity, we acknowledge that the presentation of our method can be improved and will revise the manuscript accordingly. Additionally, we will make some choices more explicit, such as the choice of reference distribution in each setup.
>
> We propose to use a log-dispersion objective on top of a standard data-based objective (forward KL):
>
> $\mathcal{L}^\theta = \lambda\_\text{data} \mathcal{L}\_{\mathrm{data}}^\theta + \lambda\_\text{LD} \mathcal{L}\_{\mathrm{LD}}^{\theta\,(p)}$
>
> This enables regularized training on fixed datasets with target energy labels, without requiring additional target evaluations. We emphasize that **this setup differs from that in Richter & Berner (2024), also for the LDR-L2 variant (p=2)**. Richter & Berner (2024) used log-variance (p=2) as a **stand-alone on-policy** objective, which means repeatedly drawing samples from the current model and evaluating the objective, which requires additional target evaluations. In contrast, we combine log-dispersion with forward KL, which allows the use of a fixed dataset and performs no additional target evaluations beyond the target energy labels in the dataset.
>
> 1) The simplest setup uses **unbiased data** from the target distribution. Both the forward KL and the log-dispersion objective are evaluated under this unbiased data distribution, i.e., the reference distribution equals the target distribution.
> 2) In experiments leveraging a **biased dataset** $\mathcal D_n^{\text{biased}}$, we have data only from a biased distribution. We first train a generator as a density estimator on this biased distribution, which allows us to obtain a dataset $\mathcal D_m^{\text{IS}}$ from the target via importance sampling.
> Our method is now applied as follows: The forward KL is evaluated using the dataset $\mathcal D_m^{\text{IS}}$. The log-dispersion objective is evaluated using a mixture reference distribution of the dataset $\mathcal D_m^{\text{IS}}$ and the original biased dataset $\mathcal D_n^{\text{biased}}$. This allows to directly incorporate the original biased data during training, improving final performance.
> 3) In the most challenging setting, we train the model **without direct access to target samples** by gradually annealing it from a simple prior toward the target distribution. Along this path, we iteratively **construct datasets by sampling from the current model and reweighting via importance sampling**. At each step, the model is trained with forward KL on the current dataset before moving to the next. We can straightforwardly apply LDR by using the current dataset for both the forward KL objective and as the reference distribution of the LD objective.
> We refer to Appendix Section B.2 and Klitzing et al. (ICLR 2026) for an extended introduction of this annealing-based framework.
>
> **Effective sample size**
> We comment on the use of the effective sample size (ESS) to assess the quality of trained Boltzmann generators. ESS is a standard metric for evaluating sampler efficiency under importance sampling (Klitzing et al., ICLR 2026; Tan et al., ICML 2025; Vaitl & Klein, NeurIPS 2025).
>
> To estimate the expectation of an observable $h(x)$, we draw samples from the generator and reweight them with importance weights $w(x)=\frac{\tilde{p}_X(x)}{q_X^\theta(x)}$:
>
> $\sum\_{n=1}^N \frac{w\left(x\_n\right)}{\sum\_{i=1}^N w\left(x\_n\right)} h\left(x\_n\right) \underset{N \rightarrow \infty}{\longrightarrow} \int h(x) p\_X(x) \mathrm{d} x$
>
> ESS quantifies how many independent samples from $p_X$ would yield the same estimator variance as importance-weighted samples from $q_X^\theta$. To approximate the ESS, we use the reverse ESS, computed from model samples. While convenient, it can be misleading if modes are missed (as noted by the reviewer), which is why we report ESS alongside NLL and other metrics.
>
> Importantly, our method improves not only ESS: Appendix E shows consistent gains across almost all metrics and experimental settings when using LDR.
>
> **Summary**
> We will revise the manuscript to make the distinction from prior work and the overall methodology clearer. Given the reviewer’s initial uncertainty around key aspects of our method, we hope these clarifications resolve the reviewer’s concerns and would appreciate a reassessment of the score.
>
> We clarify again that our method is fundamentally different from prior on-policy approaches, such as Richter & Berner (ICLR 2024): we train entirely off-policy on fixed datasets and require no additional target evaluations. This distinction is central to our contribution.
>
> LDR is a flexible regularization framework applicable across diverse settings: training with target data, with biased data, and variational training without data. We achieve state-of-the-art results across all tested benchmarks and setups.

---

> > ### Author Rebuttal · Reviewer_JwMY · 2026-04-02
> >
> > thanks for the clarifications.

---

> > > ### Author Response · Authors · 2026-04-06
> > >
> > > We are glad that our clarifications resolved the reviewer’s comments and questions, and we thank the reviewer for the thoughtful engagement during the rebuttal.
> > >
> > > We appreciate the reviewer taking the time to carefully consider our response and are pleased that the additional explanations clarified the methodology and its distinction from prior work. We will revise the manuscript accordingly to further improve clarity and accessibility, especially to the broader ML community.
> > >
> > > We further thank the reviewer for updating their overall recommendation. We would kindly ask them to also reconsider the individual scores (e.g., originality, significance), to ensure those also reflect the updated assessment.
> > >
> > > We thank the reviewer again for their time and helpful feedback.

---

### Official Review · Reviewer_mMXr · 2026-03-16

**Soundness:** 3
**Presentation:** 4
**Significance:** 3
**Originality:** 3
**Overall Recommendation:** 5
**Confidence:** 5

**Summary:**

Training Boltzmann Generators—a class of generative models (typically normalizing flows) designed to sample physical systems, as introduced by Noé et al. in 2019—using either valid empirical samples, unnormalized target energy evaluations, or a combination of both, remains an active area of research. Standard training objectives include the forward KL, reverse KL, the log-variance objective, and recent path-gradient formulations.

In this paper, the authors generalize the log-variance objective into a "log-dispersion" metric by utilizing a p-norm instead of a strict 2-norm. This allows for an $L_1$  variant that is more robust to extreme energy outliers. More importantly, the authors argue and demonstrate that **this new term can be appended to standard data-based losses (like forward KL) as a Log-Dispersion Regularizer (LDR) for off-policy training**. By demoting the log-variance/log-dispersion objective to a regularizer, they successfully adapt it for use on fixed datasets, unlocking the ability to train on cheap, biased simulation data.

The authors evaluate this approach on molecular benchmarks (Alanine Dipeptide and Hexapeptide) across both internal and Cartesian coordinate representations. Compared to baselines such as unregularized forward KL, as well as CMT, FAB, and TA-BG in variational settings, LDR demonstrates significantly improved Effective Sample Size (ESS) and a substantial reduction in the required target energy evaluations for variational training. Notably, these massive gains in sampling efficiency are achieved without degrading the Negative Log-Likelihood (NLL) metric, which actually exhibits slight improvements due to the regularizer.

**Compliance With Llm Reviewing Policy:**

Affirmed.

**Final Justification:**

The authors addressed my concerns in the rebuttal.

**Key Questions For Authors:**

The proposed Log-Dispersion Regularization (LDR) is a highly practical and effective engineering solution for improving data efficiency. However, there are a few critical limitations and omitted baselines that currently impact my assessment of the paper's technical soundness. Addressing the following questions in the rebuttal would directly and positively influence my score for the soundness of this work:

1. Comparison with Path Gradients on Molecular Systems
The method proposed here acts as a direct, zero-order competitor to the first-order Path Gradients framework for regularizing off-policy training. While I appreciate the inclusion of this baseline for the GMM and Cartesian coordinate tasks, the authors note in the main text and Appendix D.4 that they could not achieve stable training with Path Gradients on internal coordinates using Neural Spline Flows. Given the conceptual similarity of the two approaches, an empirical comparison on molecular systems is vital.

Question: Could the authors provide a comparison with Path Gradients on the molecular systems, or provide a more rigorous discussion on why a direct performance comparison is unnecessary for evaluating LDR's state-of-the-art efficiency claims?

2. Exploring Undiscovered Modes in Biased Datasets
The ability to train on biased datasets is a major strength of this work. However, the biased datasets used in the experiments were generated by seeding trajectories directly inside the main minima of the energy surface.

Question: How does LDR behave if the biased samples completely miss a critical, kinetically relevant mode? Does the two-stage refinement process inherently recover and explore undiscovered modes, or does it strictly rely on the base model's ability to interpolate probability mass into those unvisited regions? Additional discussion or a small empirical demonstration on this boundary would clarify the method's true exploratory capabilities.

3. Scalability to Real Biological Systems and Table 2 Baselines
While LDR achieves excellent Effective Sample Size (ESS) metrics on the 22-atom alanine dipeptide, the ESS drops noticeably for the 62-atom alanine hexapeptide. Furthermore, Table 2 only benchmarks the biased dataset setting on the simpler alanine dipeptide.

Question: Could the authors include the alanine hexapeptide example in the biased dataset experiments (Table 2)? Additionally, given the ESS degradation observed on a 62-atom toy system, how do the authors anticipate this framework scaling to genuinely complex, high-dimensional real-world biological systems (e.g., the ~2000-dimensional BPTI protein targeted in foundational literature)?

**Limitations:**

yes

**Strengths And Weaknesses:**

**Strengths**

- **Significance:** The paper addresses a highly relevant and persistent bottleneck in computational physics: how to efficiently train a Boltzmann Generator. By demonstrating up to an order-of-magnitude reduction in required target energy evaluations for variational training, the authors provide substantial practical utility. Crucially, the method achieves significantly improved Effective Sample Size (ESS) metrics across a wide variety of data regimes. Furthermore, the ability to **train performant models using biased, cheap-to-generate simulation data** unlocks a highly useful paradigm for practitioners who cannot afford to run unbiased Molecular Dynamics simulations.

- **Originality**: While the log-variance objective is well-established for on-policy training , the authors creatively diagnose and solve the mathematical "full support" trap that previously prevented its use on fixed datasets. By intentionally demoting this divergence to a supplementary "shape regularizer" (LDR) to be used alongside a standard data-based objective (like forward KL), they provide a highly effective combination of existing techniques. Additionally, generalizing the objective to a p-norm—specifically the $L_1$ variant —is a clever, domain-specific adaptation that successfully mitigates the exploding gradient problem caused by massive energy outliers in molecular systems

- **Soundness**: The empirical evaluation demonstrates that the LDR method functions exactly as intended. A particularly strong indicator of the method's soundness is its effect on the Negative Log-Likelihood (NLL). Because LDR is added as a regularizer to the forward KL objective, one might theoretically expect the training NLL to degrade. However, by strictly evaluating on a massive, independently generated test set, the authors demonstrate that LDR successfully prevents the model from overfitting to the training manifold. The resulting slight improvements in test NLL, combined with massive jumps in Effective Sample Size (ESS), prove the regularizer forces the model to learn a truer representation of the underlying physics.

- **Presentation**: The paper is well-structured, and the progression from the theoretical failure of off-policy log-variance to the proposed LDR solution is clearly articulated.

**Weakness**

- **Originality / Soundness (Baseline Comparisons)**: A critical weakness of this submission is the absence of a direct empirical comparison against the closest conceptual competitor: the Path Gradients method for internal coordinates. Both LDR (zero-order energy evaluations) and Path Gradients (first-order force evaluations) attempt to solve the exact same problem of off-policy data inefficiency by regularizing the forward KL objective. The authors excuse this omission by stating they could not achieve stable training with Path Gradients on Neural Spline Flows. However, since the authors also fine-tune Flow Matching models on Cartesian coordinates  (where Path Gradients have been proven stable in concurrent literature), omitting this head-to-head comparison for internal coordinates leaves the broader claims of state-of-the-art "efficiency" (compute vs. performance) somewhat incomplete.

- **Significance (Scope of Impact & Scalability)**: The authors' claims regarding "data efficiency" are somewhat overstated when looking through the lens of scalability to real-world molecular biology. While LDR achieves near-perfect Effective Sample Size (ESS) metrics (~98%) on the 22-atom alanine dipeptide , performance rapidly degrades as the system size increases. On the relatively small 62-atom alanine hexapeptide , the ESS drops to the 30% to 40% range , despite requiring massive computational budgets of up to 5 million training samples or 100 million target energy evaluations. Furthermore, **the authors noticeably omit the larger alanine hexapeptide from the biased dataset experiments in Table 2**, testing that specific setting only on the simpler dipeptide. This sharp degradation raises serious questions about the framework's true scalability. If the model's efficiency degrades this significantly on a 62-atom toy system, it is difficult to envision how it will scale to genuinely complex, high-dimensional molecular systems—such as the ~2000-dimensional BPTI protein targeted in the original  Noé et al. (2019) paper, although there are many computational tricks involved in that original paper.

**Neutral**

The following points are suggestions and comments to the authors

- **Incorrect Appendix Reference**, In Section 4.3 under the "Dataset" heading, Line 322, the authors write, "Details can be found in Appendix 6.". Appendices in this paper are lettered , so "Appendix 6" does not exist. They likely meant to reference Appendix B.4.

- **A more gradient-friendly loss alternative to $L_1$ and $L_2$** : In Appendix A.4 (Lines 722-727), the authors correctly identify that for the $L_1$ objective ( $p=1$ ), the sign function is not uniquely defined at zero. As a consequence, the gradient does not smoothly vanish at the optimum, leading to persistent gradient noise and jitter. While $L_1$ is justifiably chosen for its robustness to extreme energy outliers compared to $L_2$, this convergence issue is a well-known limitation. I highly recommend the authors explore using a Huber loss-or a Student-t distribution-as the dispersion penalty. A Huber-style Log-Dispersion Regularizer would act as an $L_1$ penalty for large energy deviations (preserving the crucial outlier robustness) while smoothly transitioning to an $L_2$ penalty for small deviations (allowing the gradients to vanish smoothly as the model reaches the optimum).

---

> ### Author Rebuttal · Authors · 2026-03-30
>
> We thank the reviewer for the positive assessment and helpful comments. We address all questions and comments below.
>
> **Comparison to path-gradient forward KL**
> We did not include a comparison to the path-gradient forward KL in internal coordinates mainly due to technical reasons: the periodic torsion degrees of the internal coordinate representation yield discontinuous gradients of the neural spline flow’s log density. Applying path-gradient forward KL in this setting amounts to regularizing a model that has discontinuous log-density gradients using continuous target gradients, leading to unstable training and poor performance. This likely cannot be avoided without changing the architecture.
>
> This **does not apply to our experiments in Cartesian coordinates**. We thus already included the path-gradient forward KL baseline for alanine dipeptide in our submitted manuscript (see Appendix, Table 13). Given the same compute budget, LDR-regularized forward KL outperforms the path-gradient-regularized forward KL by a significant margin. This addresses the “empirical comparison on molecular systems” requested by the reviewer. We further note that LDR is faster to evaluate and more flexible than path-gradient regularization, as it can be applied to arbitrary reference distributions.
>
> Finally, we do not actually view LDR as a “competitor” to path-gradient forward KL. LDR is a flexible regularization framework that can be combined with any data-based objective, including the path-gradient forward KL. Combining both yields a doubly regularized objective: path-gradient forward KL provides a first-order regularizer (target gradients), while LDR adds a zero-order regularizer (target energies). We plan to include results on this in the final version.
>
> **Scalability**
> We agree that scalability is important and will include an extended discussion in the final manuscript. We note that the effective sample size (ESS) is expected to deteriorate in high dimensions; even simple systems show exponential decay analytically. The drop in ESS in our experiments when increasing dimensionality is therefore not a limitation of our method, but a fundamental challenge of modeling Boltzmann distributions.
>
> Scaling Boltzmann generators is an active area of research, and substantial progress has been made in the past years. Methods such as annealed importance sampling (Midgley et al., 2023) or sequential Monte Carlo (Tan et al., 2025) are promising candidates for scaling Boltzmann generators. However, this is beyond our scope; we thus report standard importance-sampling ESS.
>
> We expect the combination of our LDR approach and Constrained Mass Transport (CMT) to scale particularly well to large systems. Notably, CMT enforces a constant overlap during training, smoothing the target landscape and effectively flattening importance weights. Consequently, this reduces noise in the log-dispersion objective and creates a particularly well-conditioned setting for LDR.
>
> **Exploring Undiscovered Modes in Biased Datasets**
> Since it is based on importance sampling, our self-refinement approach requires the biased distribution to cover all modes. We agree that our biased distribution for alanine dipeptide is somewhat constructed and requires chemical knowledge. However, alternative biasing methods (e.g., elevated temperature) require less prior knowledge. Since our goal was to benchmark LDR rather than biasing strategies, we chose a straightforward setup.
>
> We note that the importance-sampling approach shown for alanine dipeptide is not the only way to incorporate biased datasets. If a small number of equilibrium samples is given, one can add arbitrary biased datasets to be additionally included in the LDR objective. In this case, the biased distribution does not have to cover all modes, since no importance sampling is performed. We are currently working to include such an experiment for alanine hexapeptide in the final version.
>
> **Additional variants of the LD objective**
> We thank the reviewer for suggesting additional variants of LDR. In the final version, we will further generalize the log-dispersion objective to
>
> $\mathcal{L}\_{\mathrm{LD}}^\theta=\mathbb{E}\_{r\_X}\left[\rho\left(f^\theta(x)-c^\theta\right)\right], \quad c^\theta=\mathbb{E}\_{r\_X}\left[f^\theta(x)\right],$
>
> where $\rho(u) \geq 0$ and $\rho(u)=0 \Longleftrightarrow u=0$. This recovers LDR-Lp objectives ($\rho(u)=|u|^p$), but now also includes Huber, Student-$t$, and other dispersion penalties.
>
> We performed additional experiments on alanine hexapeptide with p=1.5, p=3, and the Huber objective: https://ibb.co/FkHJjspG. Huber and p=1.5 yield slightly improved results, while p=3 is unstable and performs worse.
>
> **Summary**
> We thank the reviewer for the valuable comments and positive assessment of our work. We have addressed all questions and concerns raised and hope that our revisions further support its strong contribution to the field.

---

> > ### Author Rebuttal · Reviewer_mMXr · 2026-04-02
> >
> > Thank you for your reply. This is a nice piece of work. I agree that the path‐gradient approach can complement the LDR you proposed, although implementing it in practice can be challenging. I’m also glad you explored a more general LD objective and achieved better results.
> >
> > I have raised my soundness score by 3 and maintained my overall score at 5.

---

> > > ### Author Response · Authors · 2026-04-06
> > >
> > > We thank the reviewer for their time and thoughtful comments during the rebuttal phase, which helped further strengthen our work. We are glad that our response, particularly regarding the comparison to the path-gradient method, scalability, and further generalization of the log-dispersion objective, addressed the reviewer’s comments to their satisfaction.
> > >
> > > We appreciate the positive assessment and are encouraged that our framework will be a useful contribution to the community.

---

### Decision · Program_Chairs · 2026-04-30

**Decision:**

Accept (regular)

**Comment:**

This paper proposes off-policy log-dispersion regularization to increase the data efficiency of Boltzmann Generators. The proposed method shows clear gains on the provided empirical results. The reviewers all ended with a positive assessement of the paper, agreeing that it presents a practical and sound contribution to this community.

Reviewers initially had questions concerning the relationship to path-gradient method, clarity of the baselines and setting, as well as scalability. These were successfully addressed during the rebuttal.

I find this paper to be a well executed paper around a simple but effective additional regularization. I recommend the paper for acceptance.